# PEBP1 amplifies mitochondrial dysfunction-induced integrated stress response

Ling Cheng[1†], Ian Meliala[1†], Yidi Kong[1], Jingyuan Chen[1], Christopher G Proud[2], Mikael Björklund[1,3]*

[1]Centre for Cellular Biology and Signalling, Zhejiang University-University of Edinburgh (ZJU-UoE) Institute, Haining, China; [2]Lifelong Health, South Australian Health & Medical Research Institute, Adelaide, Australia; [3]University of Edinburgh Medical School, Biomedical Sciences, College of Medicine & Veterinary Medicine, University of Edinburgh, Edinburgh, United Kingdom

## eLife Assessment

In this article, Cheng et al present an **important** finding that advances the understanding of mitochondrial stress response(s). The authors employed mass spectrometry-based methods in conjunction with standard molecular and cellular biology techniques to provide **compelling** evidence that phosphatidylethanolamine-binding protein 1 (PEBP1) acts as a pivotal regulator of the mitochondrial component of integrated stress response. Notwithstanding that this discovery is likely to be of significant interest to researchers across a broad spectrum of disciplines ranging from cell biology to neuroscience, it was thought that further mechanistic dissection of the role of PEBP1 in modulating integrated stress response may further strengthen this study.

**\*For correspondence:**
mikael.bjorklund.lab@gmail.com

[†]These authors contributed equally to this work

**Competing interest:** The authors declare that no competing interests exist.

**Abstract** Mitochondrial dysfunction is involved in numerous diseases and the aging process. The integrated stress response (ISR) serves as a critical adaptation mechanism to a variety of stresses, including those originating from mitochondria. By utilizing mass spectrometry-based cellular thermal shift assay (MS-CETSA), we uncovered that phosphatidylethanolamine-binding protein 1 (PEBP1), also known as Raf kinase inhibitory protein (RKIP), is thermally stabilized by stresses which induce mitochondrial ISR. Depletion of PEBP1 impaired mitochondrial ISR activation by reducing eukaryotic translation initiation factor 2α (eIF2α) phosphorylation and subsequent ISR gene expression, which was independent of PEBP1's role in inhibiting the RAF/MEK/ERK pathway. Consistently, overexpression of PEBP1 potentiated ISR activation by heme-regulated inhibitor (HRI) kinase, the principal eIF2α kinase in the mitochondrial ISR pathway. Real-time interaction analysis using luminescence complementation in live cells revealed an interaction between PEBP1 and eIF2α, which was disrupted by eIF2α S51 phosphorylation. These findings suggest a role for PEBP1 in amplifying mitochondrial stress signals, thereby facilitating an effective cellular response to mitochondrial dysfunction. Therefore, PEBP1 may be a potential therapeutic target for diseases associated with mitochondrial dysfunction.

## Introduction

Effective communication between mitochondria and the cytosol is vital for cellular and systemic homeostasis (*Quirós et al., 2016*; *Galluzzi et al., 2018*). A wide range of chronic diseases from neurodegeneration to type 2 diabetes (*Nunnari and Suomalainen, 2012*) and even many spaceflight

health risks (*da Silveira et al., 2020*) are associated with mitochondrial dysfunction. It therefore seems imperative for cells to have mechanisms to sense, integrate, and respond appropriately to mitochondrial stresses. In mammalian cells, mitochondrial dysfunction elicits an integrated stress response (ISR) in the cytosol. The main function of the ISR is to manage cellular stress and to restore homeostasis, although depending on the intensity and the duration of the stress, the ISR can also induce cell death (*Pakos-Zebrucka et al., 2016*; *Condon et al., 2021*; *Fessler et al., 2020*; *Guo et al., 2020*; *Mick et al., 2020*; *Quirós et al., 2017*; *Ryan et al., 2021*). ISR can be triggered by various insults including deprivation of amino acids or heme, ER stress, and viral infections, each of which activates one of the four eukaryotic translation initiation factor 2α (eIF2α) kinases, GCN2, heme-regulated inhibitor (HRI) kinase, PERK, and PKR respectively, leading to the phosphorylation of eIF2α, encoded by the *EIF2S1* gene (*Pakos-Zebrucka et al., 2016*). Phosphorylation of eIF2α leads to inhibition of overall translation but promotes preferential translation of specific mRNAs, such as that encoding ATF4, a transcription factor regulating redox and amino acid metabolism (*Wang and Proud, 2022*).

While the core cytoplasmic components of ISR have been known for many years, the mechanisms relaying mitochondrial stress signals to cytoplasm to induce the ISR are only beginning to be identified (*Condon et al., 2021*; *Fessler et al., 2020*; *Guo et al., 2020*; *Mick et al., 2020*). Recently, a proteolytic signaling cascade involving the mitochondrial proteins OMA1 and DELE1 and which converges on heme-regulated kinase EIF2AK1 (HRI) mediated eIF2α phosphorylation was identified (*Fessler et al., 2020*; *Guo et al., 2020*). Additional stress relay mechanisms and regulators are thought to exist as, even in the absence of DELE1, some mitochondrial stresses can still promote eIF2α phosphorylation (*Guo et al., 2020*). Further work has demonstrated that distinct mitochondrial signals activate the ISR differentially depending on the metabolic state (*Mick et al., 2020*) and with variable kinetics (*Condon et al., 2021*). Therefore, how mitochondrial dysfunction induces ISR remains an incompletely resolved question with substantial biomedical potential (*Nunnari and Suomalainen, 2012*).

Genetic screens have been instrumental in identifying genes and pathways related to mitochondrial stress signals (*Fessler et al., 2020*; *Guo et al., 2020*; *To et al., 2019*), but they may not fully capture the dynamic and complex nature of biological regulation. Mass spectrometry-based cellular thermal shift assay (MS-CETSA), also known as thermal proteome profiling (TPP), is a biophysical method that measures the thermal stability of proteins in cells or cell lysates (*Mateus et al., 2018*; *Dai et al., 2019*; *Sridharan et al., 2019*; *Perrin et al., 2020*; *Martinez Molina et al., 2013*). In this assay, cells grown under different conditions are exposed to non-physiological temperatures that induce protein unfolding and precipitation. The amount of soluble protein remaining at each temperature is quantified and used an indicator of the physiological or pathological cell state. This is because thermal stability of proteins is influenced by interactions with other proteins and ligands, such nucleic acids, lipids, and metabolites. Therefore, this method can provide complementary information to that from genetic screens, proteomics, and metabolomics. Furthermore, thermal stability can be informative about transitions between cell states in time scales where mRNA, total protein, or metabolite levels remain largely unchanged (*Dai et al., 2019*).

By combining the MS-CETSA method with multiple metabolic perturbations induced by commonly used small-molecule inhibitors, we systematically studied alterations in potential interaction states of proteins upon metabolic stress as indicated by differences in protein stability. We now report that phosphatidylethanolamine-binding protein 1 (PEBP1), also known as Raf kinase inhibitory protein (RKIP) (*Yeung et al., 1999*), is thermally stabilized upon induction mitochondrial ISR. Knockdown and knockout (KO) of PEBP1 attenuated mitochondrial ISR by reducing the phosphorylation of eIF2α while the response to endoplasmic reticulum stress induced by tunicamycin was not affected. PEBP1 interacted with eIF2α while Ser51 phosphorylation of eIF2α terminated this interaction. Our results demonstrate that PEBP1 amplifies the mitochondrial ISR thereby helping the cells to respond more robustly to mitochondrial dysfunction.

## Results
### TPP identifies proteins responding to metabolic stresses

To systematically identify alterations in thermal stability of proteins after metabolic perturbations, we treated 143B osteosarcoma cells, a common model system to study mitochondrial functionality, with several small-molecule inhibitors (*Figure 1A*). These treatments were selected to target oxidative

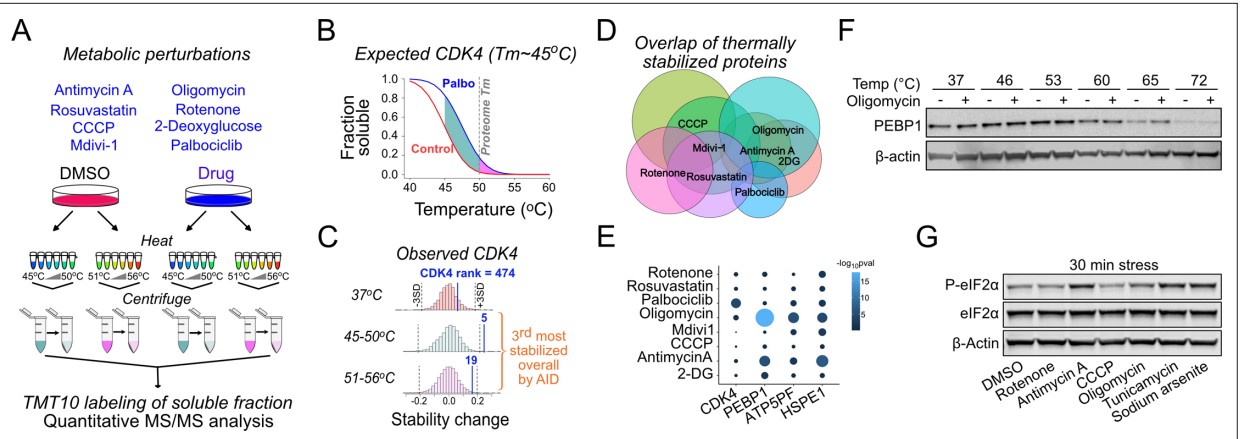

**Figure 1.** Metabolic perturbation-induced protein state changes identify PEBP1. (**A**) Schematic of thermal proteome profiling approach. Eight different drugs, including CDK4/6 inhibitor as the positive control, were used to target metabolism in 143B cells. After heating the samples at 12 different temperatures, temperatures below and above the global proteome melting temperature (50°C) were pooled, heat-denatured proteins removed my centrifugation and analyzed by mass spectrometry. Unheated (37°C) samples and DMSO solvent control were included. (**B**) Schematic showing the expected thermal denaturation curves for control and palbociclib-treated cells. The green and pink area between the curves shows the integral of the thermal shift detected as difference in CDK4 protein levels in the lower and higher temperature pool. (**C**) Histograms showing the observed CDK4 ranking in palbociclib-treated cells in the pools heated at different temperature ranges. The overall ranking was obtained using the analysis of independent differences (AID) multivariate analysis. (**D**) Venn diagram showing the overlap between thermally stabilized proteins with different drugs. A 5% false discovery rate was used to identify the hits. (**E**) Specificity of protein state alterations illustrated as AID scores (-log₁₀pval) across different drug treatments for selected hits. Note the similarity between oligomycin and antimycin A treatments. (**F**) Validation of PEBP1 thermal stabilization by oligomycin. Western blot is representative from at least three experiments. (**G**) Integrated stress response induction after a 30 min treatment with 1 μM rotenone, antimycin A, CCCP, or oligomycin as well as 5 μM tunicamycin or 25 μM sodium arsenite.

The online version of this article includes the following source data and figure supplement(s) for figure 1:

**Source data 1.** Original files for western blot analysis displayed in *Figure 1*.

**Figure supplement 1.** Further characterization of PEBP1 thermal stability in cells and using purified protein.

**Figure supplement 1—source data 1.** Original files for western blot analysis displayed in *Figure 1—figure supplement 1*.

phosphorylation (OxPhos complex (C)I=rotenone, CIII = antimycin A, CV = oligomycin, uncoupler CCCP), glucose uptake (2-deoxyglucose), mitochondrial division (Mdivi-1), or cholesterol synthesis (rosuvastatin). We also included the CDK4/6 inhibitor palbociclib as our positive control (*Miettinen et al., 2018*). For proteomics, we used an experimental strategy where samples treated at different temperatures are pooled (*Gaetani et al., 2019*), thereby compressing individual thermal melting curves into an area under the curve (AUC) quantity. We reasoned that treating intact cells with the drugs for only 30 min would allow us to observe rapid and direct effects related to metabolic flux and/ or signaling related to mitochondrial dysfunction in the absence of major changes in protein expression levels. We lysed the drug-treated cells and heated the lysates at 12 different temperatures below and above the median proteome melting temperature (Tm), which is ~50°C in human cells (*Miettinen et al., 2018*; *Jarzab et al., 2020*). We pooled the 45–50°C and 51–56°C temperature samples separately and used a non-heated lysate as a reference point for thermal stabilization (*Figure 1A*). After removing heat-denatured proteins by centrifugation, the samples were labeled with tandem mass tags (TMT10) and analyzed by mass spectrometry.

Our positive control, CDK4, was thermally stabilized upon binding of palbociclib, as expected (*Miettinen et al., 2018*; *Figure 1B*). CDK4, which has a Tm of ~45°C, ranked as the fifth most stabilized protein in the 45–50°C pool when comparing palbociclib-treated and control cells, while in the 51–56°C pool the stabilization was not quite as strong as expected (*Figure 1C*). Among the ~3700 proteins detected across all treatments, many proteins were identified as being thermally stabilized by one or more drugs, indicating overlap in the cellular responses to the different metabolic perturbations (*Figure 1D*). However, most of these changes are relatively small. To focus our analysis on the most significant and biologically relevant changes regardless of whether the Tm of each protein fell within the 45–50°C or 51–56°C temperature range, we aggregated the data from each protein into a single multivariate normal observation score using analysis of independent differences (AID; see

Materials and methods, *Supplementary file 1, table S1*). This statistical analysis ranked the proteins into the most likely thermally shifted ones using a single number. For example, CDK4 was ranked as the third most stabilized protein in palbociclib-treated cells (*Figure 1C*, *Supplementary file 1, table S1*).

## Mitochondrial ISR induction occurs concomitant with thermal stabilization of PEBP1

Having validated our proteomics approach, we focused our analysis on the oligomycin treatment as this drug stabilized multiple subunits of the intended target, the mitochondrial ATP synthase (e.g. ATP5PF, *Figure 1—figure supplement 1A*). Other proteins affected by oligomycin included BANF1, which binds DNA in an ATP-dependent manner (*Sridharan et al., 2019*), and has also identified as an oligomycin stabilized protein in a previous MS-CETA experiment (*Sun et al., 2019*), as well as mtHSP10/HSPE1, a mitochondrial heat-shock protein, suggested to be an activator of OMA1, a stress-responsive mitochondrial protease (*Yeung et al., 2023*). The top hit in the oligomycin stabilized proteome was PEBP1, an abundant cytoplasmic protein in most human tissues (based on the Protein-Atlas database; *Uhlén et al., 2015*). PEBP1, ATP5PF, and HSPE1 were also thermally stabilized by antimycin A (*Figure 1E*), but not by the other drugs.

Western blot-based thermal shift assay validated thermal stabilization of PEBP1 (*Figure 1F*). To exclude the possibility that oligomycin affects PEBP1 independently of the inhibition of ATP synthase, we used bedaquiline, a structurally unrelated complex V inhibitor. Both oligomycin and bedaquiline increased mitochondrial membrane potential and this effect was inhibited by BAM15 (used as an uncoupler) indicating inhibition of ATP synthase (*Figure 1—figure supplement 1B*). Both oligomycin and bedaquiline stabilized PEBP1 in a thermal shift assay indicating that PEBP1 is not an ATP synthase-independent off-target of oligomycin (*Figure 1—figure supplement 1C*). Furthermore, purified recombinant PEBP1 treated with oligomycin did not display thermal stabilization in an in vitro thermal shift assay (*Figure 1—figure supplement 1D*) or in lysate CETSA assay (*Figure 1—figure supplement 1E*), indicating that oligomycin does not directly bind PEBP1.

Previous work has reported that oligomycin and antimycin are the most efficient electron transport chain targeting compounds in activating ISR target genes (*Bao et al., 2016*). A 30 min treatment with antimycin A or oligomycin that stabilized PEBP1 was also sufficient to induce eIF2α phosphorylation, the core event in the ISR (*Pakos-Zebrucka et al., 2016*; *Figure 1G*). Tunicamycin and sodium arsenite, used as positive controls, induced robust eIF2α phosphorylation. Rotenone increased eIF2α phosphorylation only after 4 hr while CCCP at the concentration used did not do so noticeably even at that time (*Figure 1—figure supplement 1F*), demonstrating that the mitochondrial ISR kinetics and intensity varies depending on the inducer (*Condon et al., 2021*; *Mick et al., 2020*). Therefore, this data suggested that stabilization of PEBP1 occurs concomitant with induction of the mitochondrial ISR (see *Figure 1—figure supplement 1G* for summary of drugs used to target PEBP1 and ISR in this manuscript).

## Loss of PEBP1 attenuates mitochondrial ISR gene expression in cultured cells and in vivo

PEBP1 has not been previously linked to mitochondrial dysfunction in mammalian cells. We performed RNA sequencing (RNAseq) with non-targeting sgRNA control KO and PEBP1 sgRNA KO cells treated with and without oligomycin for 6 hr. The first dimension of the principal component analysis (PCA) separated the control KO and PEBP1 KO cells (*Figure 2A*), PEBP1 being one of the top genes with negative PCA loadings (*Figure 2—figure supplement 1A*). The transcriptional response to oligomycin observed in control KO cells was attenuated in the PEBP1 KO cells as seen collectively in the second dimension of the PCA plot (y axis) and by single genes in a heatmap and a volcano plot (*Figure 2B and C*). Top PCA loadings (*Figure 2—figure supplement 1A*) and the volcano plot (*Figure 2C*) indicated that oligomycin induced the ISR in control KO cells as expected (*Pakos-Zebrucka et al., 2016*; *Condon et al., 2021*; *Fessler et al., 2020*; *Guo et al., 2020*; *Mick et al., 2020*; *Quirós et al., 2017*; *Ryan et al., 2021*).

While known ISR-induced genes including TRIB3, ASNS, ATF4, CEBPG, DDIT3/CHOP, and DDIT4 were upregulated by oligomycin in control cells, PEBP1 KO cells displayed a highly attenuated ISR response at the RNA level. In control cells, gene ontology (GO) analysis indicated enrichment of

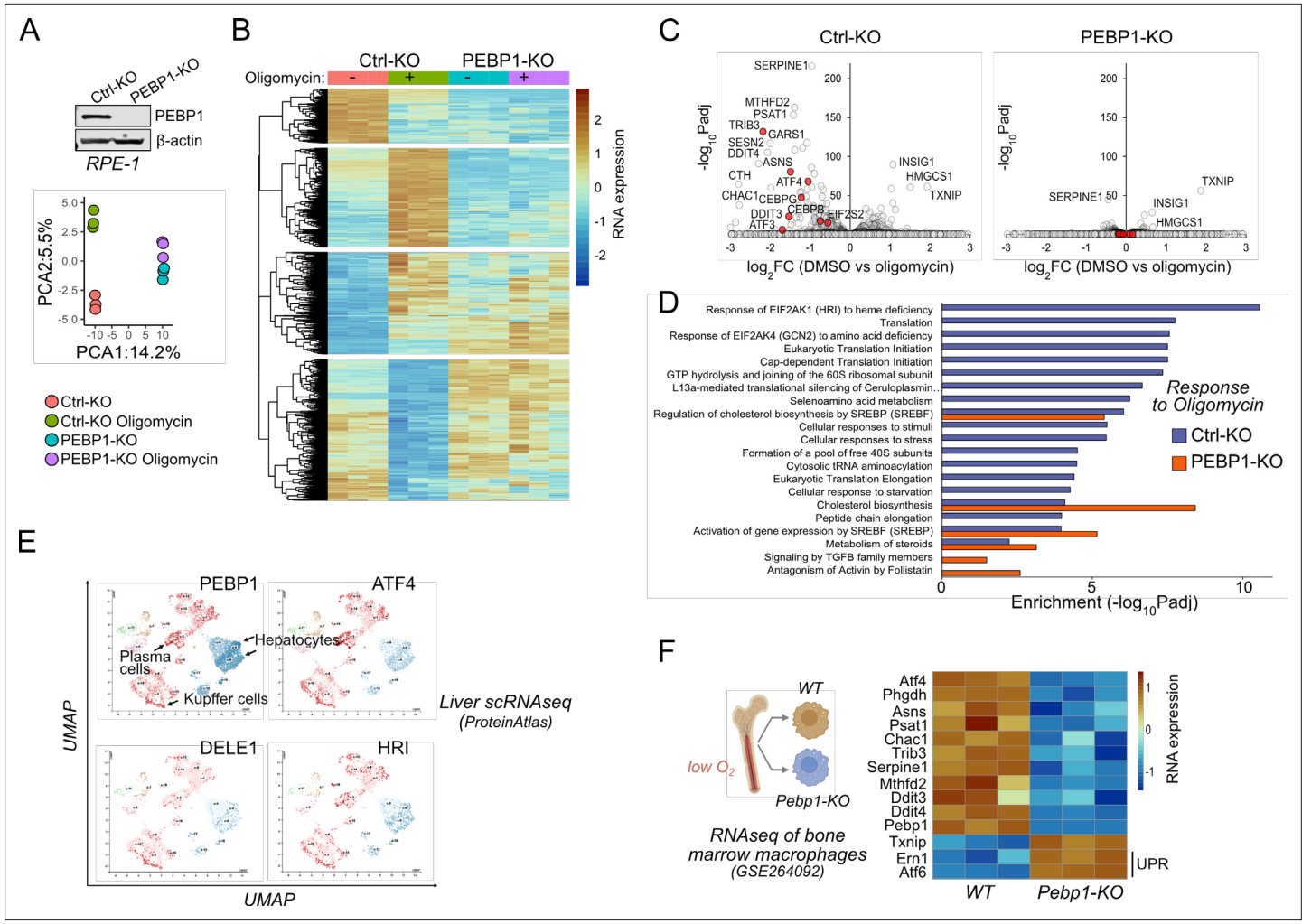

**Figure 2.** Loss of PEBP1 attenuates integrated stress response (ISR) gene expression. (**A**) PEBP1 expression in control and PEBP1 knockout (KO) cells as well as principal component analysis of RNAseq samples treated with and without 1 μm oligomycin for 6 hr. Three replicates were used for each group. (**B**) Heatmap of significantly changing genes in RNAseq. (**C**) Comparison of oligomycin-induced gene expression effects using volcano plots of Ctrl and PEBP1 KO cells. ISR genes are highlighted in red. (**D**) Gene ontology analysis of oligomycin-induced responses in control KO cells (blue) and PEBP1 KO cell (red). (**E**) Comparison of PEBP1 and mitochondrial ISR signaling component (ATF4, DELE1, HRI) expression in single-cell RNAseq from liver based on publicly available UMAP analysis from ProteinAtlas. For the names of the cell types in other clusters, see here. (**F**) ISR gene expression from public RNAseq data of bone marrow macrophages isolated from WT and *Pebp1* KO mice. Two unfolded protein response (UPR) genes *Ern1* and *Atf6* are also shown. HRI, heme-regulated inhibitor kinase.

The online version of this article includes the following source data and figure supplement(s) for figure 2:

**Source data 1.** Original files for western blot analysis displayed in **Figure 2A**.

**Figure supplement 1.** Integrated stress response (ISR) induction in control and PEBP1 knockout (KO) cells by RNAseq.

groups such as 'Response of EIF2AK1 (HRI) to heme deficiency' and 'Eukaryotic Translation Initiation', consistent with activation of HRI-mediated mitochondrial ISR. These GO groups were not enriched in PEBP1 KO cells (**Figure 2D**). In contrast, thioredoxin interacting protein (TXNIP) as well as genes involved in cholesterol biosynthesis (e.g. INSIG1, HMGCS1), which are unrelated to ISR and known to be repressed by oligomycin and other OxPhos inhibitors (**Mick et al., 2020**), remained downregulated in both control and PEBP1 KO cells (**Figure 2C**). Therefore, the ability of PEBP1 KO to attenuate expression of ISR genes but not cholesterol biosynthesis genes suggested that PEBP1 might be specifically involved in ISR signaling.

Consistent with the possibility of PEBP1 being involved in ISR signaling, single-cell RNAseq data from ProteinAtlas database suggested that PEBP1 is expressed similarly to ATF4, HRI, and DELE1 in the liver, particularly in hepatocytes, plasma cells, and Kupffer cells (**Figure 2E**). We also analyzed a

publicly available RNAseq dataset (GSE264092) of bone marrow macrophages from WT and *Pebp1* KO mice. Bone marrow is a naturally occurring low-oxygen environment (*Spencer et al., 2014*) where mitochondrial respiration may become compromised. Gene set enrichment analysis showed that there was a tendency of OxPhos genes to be downregulated in *Pebp1* KO bone marrow macrophages (*Figure 2—figure supplement 1B*). The cells from *Pebp1* KO animals displayed reduced expression of common ISR genes (*Figure 2F*), while there was a mild upregulation of unfolded protein response genes *Ern1 (Ire1α)* and *Atf6*. This gene expression data therefore suggests that the reduced expression of common ISR genes is less likely to be mediated by changes in PERK, the third UPR arm, and more likely due to suppression of ISR by *Pebp1* KO in vivo.

## PEBP1 amplifies the intensity of mitochondrial but not ER stress signaling

Western blotting indicated reduced phosphorylation of eIF2α in RPE1 cells lacking PEBP1 (*Figure 3A*). The reduction of eIF2α phosphorylation by loss of PEBP1 was observed regardless of the approach used to activate mitochondrial ISR by targeting ATP synthase as shown by the effects of treatments with oligomycin, bedaquiline, and ATP5F1A siRNA (*Figure 3—figure supplement 1*). Treatment of cells with the MEK inhibitor trametinib did not rescue eIF2α phosphorylation (*Figure 3A*) or ISR target gene activation (*Figure 3—figure supplement 2A and B*) although PEBP1 KO cells displayed increased ERK activation consistent with the role of PEBP1 as a RAF kinase inhibitor. Therefore, PEBP1 affects the mitochondrial ISR independently of its role in the RAF/MEK/ERK pathway (*Yeung et al., 1999*).

We next tested if expression of PEBP1 amplifies ISR signaling by using a luciferase-based ATF4 reporter assay (*Figure 3B*). Since PEBP1 is an abundant protein, we performed these experiments in PEBP1 KO cells as its overexpression in WT cells leads to only a minor increase in total cellular PEBP1 levels. This assay demonstrated that although PEBP1 is not necessary for HRI-mediated ATF4 reporter activation, it can amplify this HRI-induced signal. In addition, PEBP1 itself is not sufficient to induce reporter activity in the absence of HRI co-expression indicating that it acts as an auxiliary protein in the ISR signal transduction. PEBP1 S153D mutant, which mimics the PKC-catalyzed phosphorylation event that converts PEBP1 from a RAF inhibitor to an inhibitor of G protein-coupled receptor kinase 2 (GRK2) inhibitor (*Skinner et al., 2017*), slightly increased reporter activity, while P112E mutant which abolishes the interaction with lipooxygenases in ferroptosis (*Wenzel et al., 2017*) activated the reporter similarly to WT PEBP1. As a control, DELE1 lacking the mitochondria targeting sequence (DELE1-ΔMTS) which can directly interact with HRI in the cytoplasm to activate ISR (*Fessler et al., 2020*) had similar effect to WT PEBP1.

Since ISR activation involves eIF2α phosphorylation and consequent downregulation of global protein synthesis, we used incorporation of the methionine analog L-homopropargylglycine (HPG) to assess the effect of PEBP1 on protein synthesis as the commonly used puromycin assay is not reliable in energy-starved cells (*Marciano et al., 2018*). KO of PEBP1 partially rescued the rate of protein synthesis in oligomycin-treated cells but had no effect in tunicamycin-treated cells (*Figure 3C*). Taken together, the effects of PEBP1 KO on eIF2α phosphorylation, PEBP1 expression on ATF4 reporter activity, and the rescue of global protein synthesis indicate that PEBP1 influences mitochondrial ISR.

The intensity and the duration of the stress determine the adaptive and cell death inducing outcomes of the ISR (*Batjargal et al., 2023*). To test whether PEBP1 affects the amplitude or the kinetics of the stress response, we performed a time-course analysis. The phosphorylation of eIF2α induced by oligomycin was attenuated in PEBP1 KO cells at all time points (*Figure 3D*) and concentrations (*Figure 3—figure supplement 2C*), indicating that PEBP1 alters the amplitude; however, it did not affect the kinetics or the duration of mitochondrial stress signaling. Since the dynamic range of P-eIF2α immunoblotting is limited (*Krzyzosiak et al., 2022*), we also checked expression of ATF4 and its transcriptional target CHOP, both of which were increased by oligomycin in control, but not in PEBP1 KO cells (*Figure 3D*). Phosphorylation of eIF2α induced by deferoxamine mesylate (DFOM), which chelates iron and induces the ISR via the OMA1-DELE1-HRI pathway (*Sekine et al., 2023*), was also attenuated in PEBP1 KO cells (*Figure 3E* and *Figure 3—figure supplement 2D*). Similar results were obtained by stress induced by complex I inhibitor rotenone as well as viral nucleic acid mimicking poly(I:C), which activates ISR via the kinase PKR (*Figure 3—figure supplement 2E and F*). However, consistent with the protein synthesis assay, the phosphorylation of eIF2α induced by tunicamycin, a strong inducer

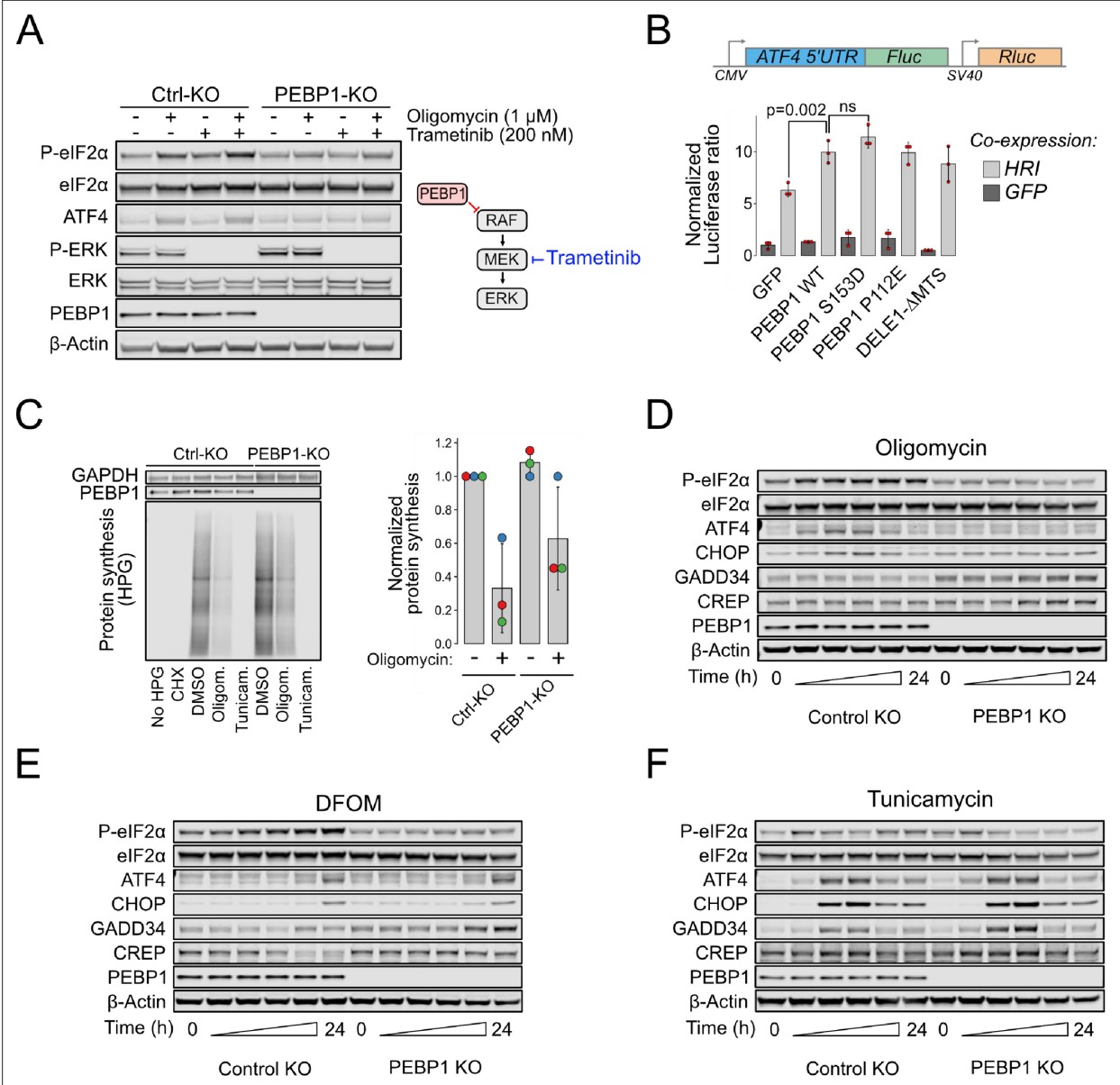

**Figure 3.** Loss of PEBP1 attenuates heme-regulated inhibitor (HRI)-mediated mitochondrial signaling but not PERK-mediated ER stress. (**A**) Western blot analysis of integrated stress response (ISR) activation in RPE1 control and PEBP1 knockout (KO) cells treated with and without MEK inhibitor trametinib. (**B**) ATF4 luciferase reporter assay in PEBP1 KO cells. Cells were transfected with the dual luciferase reporter (upper panel) in the presence or absence of the indicated genes (either co-expressing with GFP or HRI). Data shown in mean ± SD, n=3. Statistical analysis by ANOVA followed by Tukey's post hoc test. (**C**) Effect of PEBP1 on protein synthesis assayed by incorporation of a methionine analog, L-homopropargylglycine (HPG), followed by a Click-reaction with fluorescent Azide-647. Cells were treated with 1 μM cycloheximide (CHX), 5 μM oligomycin, or 5 μM tunicamycin to inhibit protein synthesis. Right panel shows quantification of oligomycin effect from three biological replicates (individual experiments highlighted with dot color). (**D**) Time-course analysis of ISR activation in RPE1 control and PEBP1 KO cells treated with 1 μM oligomycin. (**E**) Time-course of cells treated with 1 mM deferoxamine mesylate (DFOM). (**F**) Time-course of cells treated with 1 μM tunicamycin. The specific time points in panels D–F are 0, 2, 4, 8, 16, and 24 hr. Western blots are representative examples from two to three experiments.

The online version of this article includes the following source data and figure supplement(s) for figure 3:

**Source data 1.** Original files for western blot analysis displayed in *Figure 3*.

**Figure supplement 1.** Integrated stress response (ISR) induction in control and PEBP1 knockout (KO) cells is independent of the stress inducer.

**Figure supplement 1—source data 1.** Original files for western blot analysis displayed in *Figure 3—figure supplement 1*.

**Figure supplement 2.** Characterization of PEBP1-mediated integrated stress response (ISR).

**Figure supplement 2—source data 1.** Original files for western blot analysis displayed in *Figure 3—figure supplement 2*.

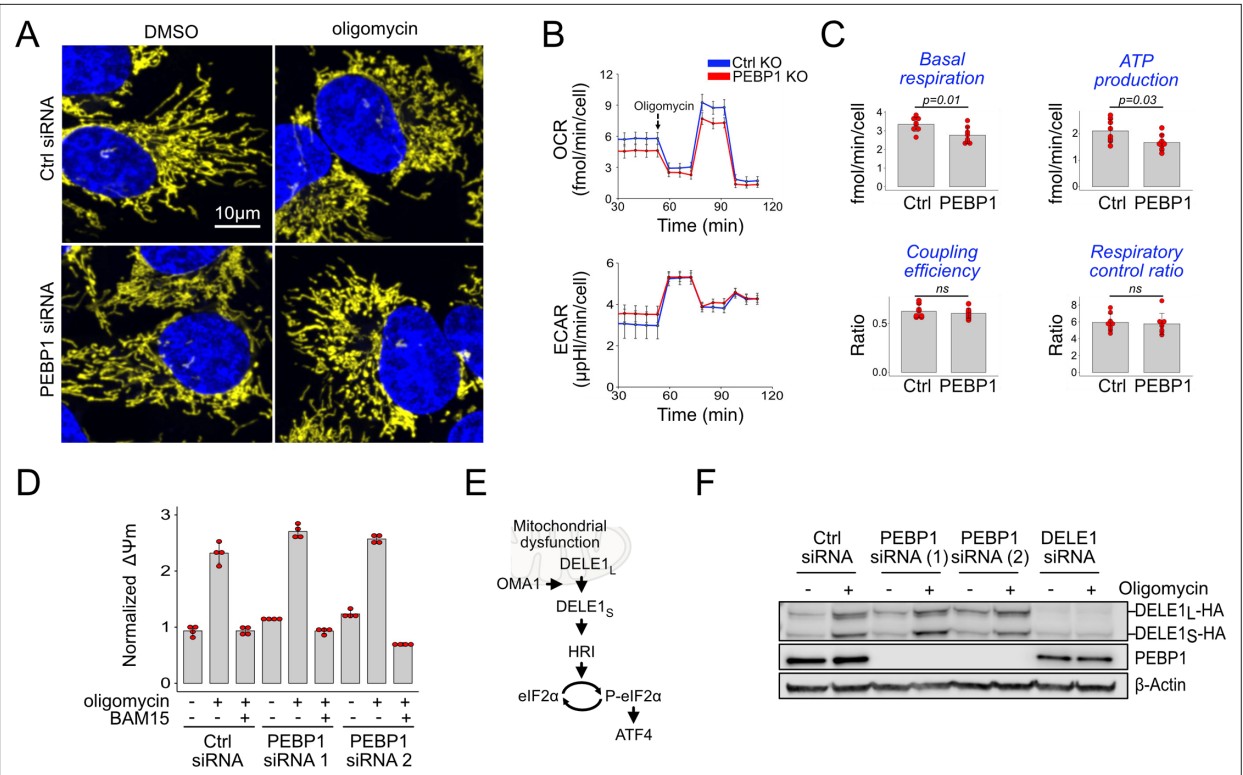

**Figure 4.** PEBP1 does not interfere with mitochondrial OMA1-DELE1 signaling. (**A**) A representative example of mitochondrial fragmentation in 143B cells treated with or without PEBP1 siRNA and 1 μM oligomycin. Mitochondria are shown in yellow, DNA in blue. (**B**) Seahorse oxygen consumption rate (OCR) and extracellular acidification rate (ECAR) analysis in RPE1 knockout (KO) cells. (**C**) Quantification of the respiratory parameters from the Seahorse analysis. Data shown in B and C is mean ± SD, n=7. The p-value is from a two-sided t test. (**D**) Mitochondrial membrane potential after treating control and PEBP1 silenced 143B cells with 1 μM oligomycin or oligomycin with 0.5 μM uncoupler BAM15. (**E**) Schematic of mitochondrial stress signal transduction. (**F**) Activation of DELE1-HA in 143B endogenous knock-in cells treated with PEBP1 and DELE1 siRNAs. The long uncleaved form (DELE1$_L$-HA) and the proteolytically processed short form (DELE1$_S$-HA) are indicated.

The online version of this article includes the following source data for figure 4:

**Source data 1.** Original files for western blot analysis displayed in *Figure 4*.

---

of ER stress, was not prevented in PEBP1 KO cells (*Figure 3F*, *Figure 3—figure supplement 2G*). Note that in PEBP1 KO cells there was a small increase in the expression of the stress-inducible eIF2α phosphatase subunit GADD34 (PPP1R15A) as well as the constitutively expressed PPP1R15B subunit. However, this increase in phosphatase levels could not explain the reduction of eIF2α phosphorylation in PEBP1 KO cells as Sephin-1 and Raphin-1, inhibitors of PPP1R15A and PPP1R15B, respectively, did not rescue eIF2α phosphorylation (*Figure 3—figure supplement 2H*). Overall, these results indicate that PEBP1 is involved in amplifying the ISR to a variety of stimuli, particularly those that induce mitochondrial dysfunction.

## Loss of PEBP1 interferes with mitochondrial stress signaling without affecting OMA1 or DELE1

We next sought to understand how PEBP1 is linked to mito-ISR signaling mediated by the OMA1-DELE1-HRI pathway (*Fessler et al., 2020*; *Guo et al., 2020*). Activation of OMA1 leads to mitochondrial fragmentation (*Baker et al., 2014*). Oligomycin fragmented mitochondria in both control and PEBP1 KO cells (*Figure 4A*). The mitochondrial oxygen consumption rate (OCR) was also reduced after oligomycin addition in both control and PEBP1 KO cells. Compared to control cells, we observed only a minor reduction in basal respiration and oligomycin-sensitive ATP production in PEBP1 KO cells (*Figure 4B and C*). Mitochondrial coupling efficiency and the respiratory control ratio were also similar to control cells. There was no increase in the extracellular acidification rate (ECAR), a parameter used

to measure glycolytic activity. Overall, this data indicates that PEBP1 does not have a major effect on metabolic activity in these cells.

Previous work identified that mitochondrial hyperpolarization mediates oligomycin-induced ISR signaling (*Mick et al., 2020*). TMRE staining confirmed that mitochondrial membrane potential was increased with oligomycin treatment, but this hyperpolarization was not reduced by PEBP1 knockdown (*Figure 4D*), while the uncoupler BAM15 readily depolarized mitochondria. We then checked whether PEBP1 influences activation of DELE1 (*Figure 4E*). Upon mitochondrial stress, OMA1 proteolytically cleaves and activates mitochondria-localized DELE1. DELE1-HA knock-in RPE1 cells showed the expected cleavage of DELE1 in response to oligomycin, but this cleavage was not influenced by PEBP1 knockdown (*Figure 4F*).

## Phosphorylation modulates interaction of PEBP1 with eIF2α

PEBP1 is thought to act as a scaffold protein for regulating cellular signaling (*Wenzel et al., 2017*; *Burack and Shaw, 2000*). Scaffolds are typically used for spatial organization of signaling components to achieve efficient and specific signal transduction (*Burack and Shaw, 2000*; *Good et al., 2011*). For example, activation of the eIF2α kinase GCN2 requires GCN1 and GCN20 as accessory proteins (*Wang and Proud, 2022*) and these proteins can be considered as scaffolds. It was also recently suggested that DELE1 oligomerization could serve as a scaffold for HRI activation (*Yang et al., 2023*).

Scaffold proteins typically function either stoichiometrically (*Heinrich et al., 2002*; *Levchenko et al., 2000*) or in excess relative to the partner kinase to allow signal amplification (*Witzel et al., 2012*). Using data from mass spectrometry-based quantification in HeLa cells (*Itzhak et al., 2016*), roughly equimolar concentrations of PEBP1 and eIF2α were observed. Instead, PEBP1 is ~10–1000 times more abundant than any of the ISR kinases and ~6000 times more abundant than DELE1 in HeLa cells (*Figure 5A*). Publicly available spatial transcriptomics data (*Chen et al., 2022*) show that PEBP1 mRNA is highly expressed in most if not all mouse tissues, particularly in the brain (*Figure 5—figure supplement 1A*). Given the abundances of the mitochondrial ISR signaling components and the observed reduction in eIF2α phosphorylation in PEBP1 KO cells, we reasoned that among the mitochondrial ISR signaling components, PEBP1 might most likely interact with eIF2α. However, co-immunoprecipitation assays failed to show an interaction between these proteins (data not shown). Since many protein interactions may be too weak or transient for detection by co-immunoprecipitation assay, we used NanoBiT luminescence complementation assay where a pair of interacting proteins brings the large (LgBiT) and small (SmBiT) luciferase fragments together to catalyze bioluminescence (*Dixon et al., 2016*). HEK293T cells expressing PEBP1-LgBiT displayed luminescence when co-transfected with eIF2α-SmBiT but not with an empty SmBiT vector (*Figure 5B*). Phorbol ester (PMA) treatment, which leads to phosphorylation of PEBP1 at Ser153, enhanced luminescence in cells transfected with WT PEBP1, but not in cells transfected with a non-phosphorylatable S153A PEBP1. This suggested a specific interaction between PEBP1-eIF2α which is enhanced by PEBP1-S153 phosphorylation. This data is consistent with the modest increase of ISR signaling by PEBP1 S153D in the ATF4 reporter assay. As a control, WB analysis demonstrated S153 phosphorylation and that expression levels of the PEBP1 and eIF2α remained unchanged by the PMA treatment (*Figure 5B*). The phosphomimetic S153D PEBP1 mutant also displayed an increased interaction with eIF2α further supporting a specific interaction between PEBP1 and eIF2α (*Figure 5—figure supplement 1B*). While these point mutations suggested specificity, we wanted to further demonstrate the effect of PEBP1 S153 phosphorylation on the interaction between PEBP1 and eIF2α. We therefore monitored bioluminescence changes in 5 min intervals upon drug-induced phosphorylation (*Figure 5C*). PMA treatment of cells transfected with WT PEBP1 but not with the S153A mutant displayed increased luminescence (*Figure 5D*).

We then tested if phosphorylation of S51 in eIF2α affects the interaction between PEBP1 and eIF2α. A phosphomimic S51D but not an S51A point mutant abolished luminescence signal suggesting that the interaction between PEBP1 and eIF2α could be terminated by S51 phosphorylation of eIF2α (*Figure 5E*). However, this low luminescence signal was explained by loss of PEBP1-LgBiT expression as S51D eIF2α-SmBiT expression likely mimicked ISR shutting down protein synthesis. To resolve this, we performed real-time assay of PEBP1 binding to WT and S51A mutant eIF2α after inducing ISR-mediated eIF2α phosphorylation with sodium arsenite. Compared to DMSO-treated controls, arsenite treatment decreased the luminescence in cells expressing PEBP1 with WT eIF2α but not with the S51A mutant (*Figure 5F*). The lack of effect in the S51A mutant suggests that lack of S51

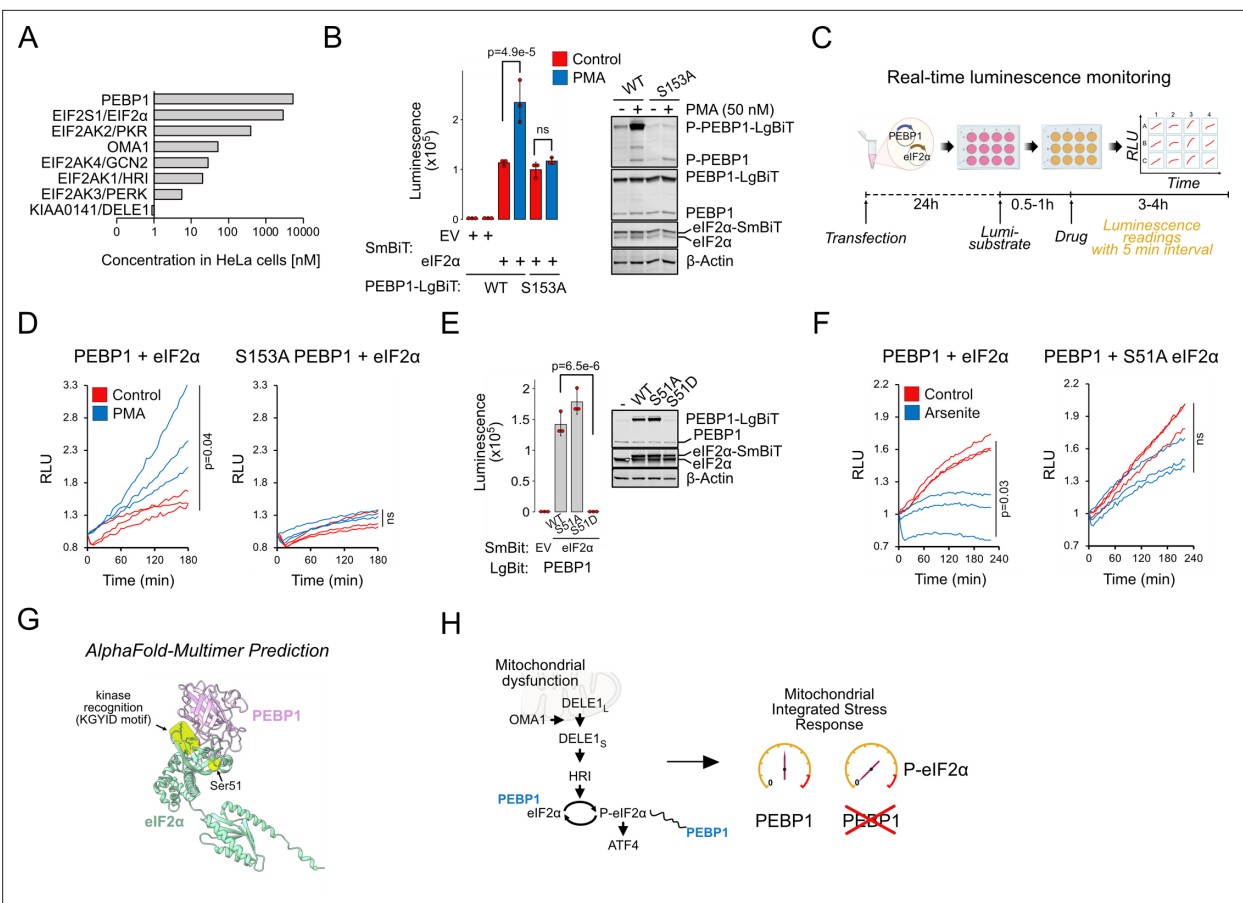

**Figure 5.** Phosphorylation modulates interaction of PEBP1 with eukaryotic translation initiation factor 2α (eIF2α). (**A**) Protein concentrations for PEBP1 and integrated stress response (ISR) components in HeLa cells based on *Itzhak et al., 2016*. Note the log10 scale. (**B**) Nanobit interaction assay for PEBP1 and eIF2α in 293T cells. Cells were transfected with the indicated constructs; after 24 hr, cells were treated with and without 50 nM PMA for 4 hr before adding luciferase substrate. Right panel shows the expression levels in cells transfected with WT and S153A PEBP1-LgBiT. Data is representative of three independent biological experiments. (**C**) Schematic of the real-time interaction monitoring. (**D**) Real-time analysis of PEBP1 phosphorylation-dependent interaction with eIF2α using WT and S153A PEBP1. Cells were treated with 50 nM PMA. The red and blue traces show data from individual wells, n=3. (**E**) Effect of eIF2α-S51 mutations on PEBP1-eIF2α interaction. Western blot (WB) (right) shows expression of endogenous and exogenous eIF2α and PEBP1 levels. (**F**) Real-time analysis of PEBP1 interaction with WT and S51A eIF2α. Cells were treated with 25 μM sodium arsenite. Data is representative of three independent biological experiments with three replicate wells in each experiment. (**G**) AlphaFold prediction of the putative complex structure between PEBP1 and eIF2α. The KGYID kinase recognition motif and eIF2α-Ser51 location are indicated. (**G**) Schematic of PEBP1 in mediating mitochondrial ISR amplification. For panels B and E, data shown is mean ± SD, n=3. Statistical analysis by ANOVA, followed by Tukey's post hoc test. For panels D and F, area under curves were calculated and significance tested with two-sided t test.

The online version of this article includes the following source data and figure supplement(s) for figure 5:

**Source data 1.** Original files for western blot analysis displayed in *Figure 5*.

**Figure supplement 1.** PEBP1 and eukaryotic translation initiation factor 2α (eIF2α) expression and interaction.

**Figure supplement 1—source data 1.** Original files for western blot analysis displayed in *Figure 5—figure supplement 1*.

phosphorylation of eIF2α is indeed critical for weaking the interaction between PEBP1 and eIF2α. The S51 phosphorylation-dependent loss of protein-protein interaction is consistent with thermal stabilization of PEBP1 upon mitochondrial stress as proteins in larger complexes are generally less thermally stable. Finally, AlphaFold-Multimer analysis predicted that PEBP1 may bind eIF2α close to the S51 and the KGYID kinase recognition motif of eIF2α (*Dey et al., 2005*; *Figure 5G*), suggesting a structural basis for how PEBP1 could influence the phosphorylation of eIF2α. Taken together, these results indicate that PEBP1 acts as an amplifier of the mitochondrial ISR by interacting with eIF2α (*Figure 5H*).

## Discussion

How the intensity and duration of a stress determine the cellular stress response is complex and remains incompletely characterized. Here, we have identified that PEBP1 amplifies the ISR following mitochondrial dysfunction, but not other inducers of the ISR. Loss of PEBP1 reduced the cytoplasmic ISR as measured by eIF2α phosphorylation upon mitochondrial stress induced by oligomycin or iron chelation, both of which are mediated by the OMA1-DELE1-HRI pathway (*Sekine et al., 2023*). Importantly, PEBP1 depletion did not directly affect OMA1-DELE1-HRI pathway or repression by oligomycin of genes involved in the cholesterol synthesis pathway (*Mick et al., 2020*), indicating that PEBP1 is not a general mediator of the effects of OxPhos inhibition. Instead, PEBP1 dynamically interacted with eIF2α suggesting a direct and specific role in modulating ISR.

Our findings raise the question 'why do cells utilize an additional protein, PEBP1, for mitochondrial ISR activation?'. PEBP1 did not influence tunicamycin induced stress, which is mediated by the eIF2α kinase PERK. Compared to tunicamycin, mitochondrial stress signals seem to be relatively weak activators of the ISR. For example, at least in the cell types used here, oligomycin caused a less severe inhibition of global protein synthesis and weaker ATF4 induction than tunicamycin. This may reflect differential needs for stress management, where it may be more important to handle mitochondrial stress through HRI-mediated mitophagy which does not require ATF4 (*Chakrabarty et al., 2024*). Therefore, we propose that PEBP1 may be required for amplifying weak stress signals, especially those that originate from mitochondria. Quantitative comparison of the signaling component amounts in HeLa cells (*Figure 5A*) illustrates the need for substantial amplification of mitochondrial stress signals. However, efficient signal amplification should not respond to 'noise'. It was previously suggested that additional regulation of OMA1-DELE1-HRI pathway may prevent accidental stress response induction caused by proteolytically processed cytoplasmic DELE1 (*Guo et al., 2020*). Signal amplification by PEBP1 may in part ensure that ISR activation results from authentic mitochondrial dysfunction. Overall, our results indicate that an efficient relay of mitochondrial stress to the cytosol via the OMA1-DELE1-HRI pathway requires PEBP1, at least in some cell types.

Mechanistically, luminescence complementation assays indicated that PEBP1 interacts with eIF2α, and that this interaction is reduced upon stress-induced phosphorylation of eIF2α-S51. Although our interaction data cannot explain the increased phosphorylation of eIF2α in cells expressing PEBP1, the dynamic interaction with eIF2α and the abundance of PEBP1 relative to other components of the mitochondrial ISR pathway suggests that PEBP1 may function as a scaffold protein for efficient signal transduction and amplification (*Figure 5H*). The current evidence, based on the observations that S153D only slightly increased ATF4 reporter activity and interaction with eIF2α in the luminescence complementation assay, suggests that S153 phosphorylation of PEBP1 is not critical for amplification of mitochondrial ISR. This effectively rules out the main previously indicated mechanism of specificity in PEBP1 signaling, where S153 phosphorylation converts PEBP1 from a RAF inhibitor to a GRK2 inhibitor (*Skinner et al., 2017*). Consistently, induction of ISR with oligomycin, arsenite, or tunicamycin did not induce notable S153 phosphorylation based on WB analysis with the phospho-specific PEBP1 antibody (data not shown).

The identified role for PEBP1 in ISR may have broad biomedical relevance as small-molecule ISR inhibitors may potentially increase both healthspan and lifespan (*Derisbourg et al., 2021*). Enhancing or prolonging phosphorylation of eIF2α using small molecules is now being explored for disorders associated with dysregulated ISR signaling (*Pakos-Zebrucka et al., 2016*). For example, ISR activation is thought to play a causative role in a wide array of cognitive and neurodegenerative disorders (*Costa-Mattioli and Walter, 2020*). Amplification of mitochondrial stress signals may be particularly important in brain and other tissues with high metabolic demands and vulnerability to mitochondrial dysfunction. PEBP1 is particularly abundant in the brain, where it is known as the precursor of hippocampal cholinergic neurostimulating peptide (*George et al., 2006*). Given the crucial role of the ISR in the hippocampus for memory formation (*Costa-Mattioli and Walter, 2020*; *Sharma et al., 2020*), it will be important to determine whether PEBP1 plays a role in these processes and thus whether PEBP1 could be targeted therapeutically in any age-related cognitive or neurodegenerative disorders.

Our study has several limitations. The use of specific cell lines and the lack of functional data from in vivo experiments limit the generalizability of these findings. Apart from the detected interaction with eIF2α, our analyses are not able to pinpoint the precise mechanism by which PEBP1 enhances eIF2α phosphorylation in response to mitochondrial stresses, but not during tunicamycin-induced ER stress.

While the eIF2α kinases are differently located in the cell - PERK on the ER, GCN2 on ribosomes, HRI cytoplasmic, it remains possible that there are also separate pools of eIF2α such as that present on mitochondria as recently identified (*Chakrabarty et al., 2024*). This remains an important point for future research.

In conclusion, by identifying PEBP1 as an amplifier of mitochondrial stress signals, our study provides new insights into cytoplasmic mechanisms which ensure appropriate responses to acute mitochondrial dysfunction. These findings may have important implications for understanding mitochondrial diseases and in developing new therapeutic strategies.

## Materials and methods

### Cell culture

Human cell lines 143B (CRL-8303; RRID:CVCL_2270) and hTERT RPE-1 (CRL-4000; RRID:CVCL_4388) were obtained from ATCC. Human HEK293T (RRID:CVCL_0063) cells were from GeneCopoeia, Guangzhou, China. 143B, RPE1, and HEK293T cells were cultured in 10% FBS (Cat.No. VIS368948, VisTech) in DMEM containing pyruvate and L-alanyl-L-glutamine (FI101-01, Transgenbiotech, Beijing). All cell lines were cultured in the presence of penicillin and streptomycin and tested negative for mycoplasma.

### siRNA silencing

For siRNA knockdown, 143B or RPE1 cells (50,000 cells/ml) were plated in six-well plates (2 ml/well) the day before transfection. 25 nM SiRNA was transfected using Lipofectamine 3000 (L3000-015, Thermo Fisher) prepared in OptiMEM medium (Thermo Fisher). After 48 hr, cells (60,000 cells/ml) were reseeded into six-well plates (2 ml/well) for drug treatment. The following sequences were synthesized by Tsingke (Beijing) and used for the experiments. A proprietary non-targeting siRNA (Cat.No. TSGXS101) from Tsingke was used as a negative control.

### Oligonucleotides

siRNA sequences:

| | Forward primer sequence (5'- 3') | Reverse primer sequence (5'-3') |
|---|---|---|
| siDELE1-1 | CAGCGACCUGACAGUUACA(dT)(dT) | UGUAACUGUCAGGUCGCUG(dT)(dT) |
| siPEBP1-1 | GCACAGUCCUCUCCGAUUA(dT)(dT) | UAAUCGGAGAGGACUGUGC(dT)(dT) |
| siPEBP1-2 | GAUUCAGGGAAGCUCUACA(dT)(dT) | UGUAGAGCUUCCCUGAAUC(dT)(dT) |
| siATP5F1A-1 | GAUCAUCUAUGACGACUUA(dT)(dT) | UAAGUCGUCAUAGAUGAUC(dT)(dT) |

Sequences for generating KO cell lines:

| | |
|---|---|
| PEBP1 KO sgRNA1 | GCATGTCACCTACGCCGGGGCGG |
| PEBP1 KO sgRNA2 | CCCGGATGCTCCCAGCAGGAAGG |
| Non-targeting control sgRNA | TGAGCATTCGTAGCCCAGCA |

Sequences for endogenous HA tagging of DELE1:

| | |
|---|---|
| DELE1 KI sgRNA | GAAAGGAGTGTTGTAAGACTAGG |
| ATP1A1 sgRNA | GAGTTCTGTAATTCAGCATA |

The qRT-PCR primers used were the following:

| Gene | Forward primer sequence (5'- 3') | Reverse primer sequence (5'-3') |
|---|---|---|
| DDIT4 | TGAGGATGAACACTTGTGTGC | CCAACTGGCTAGGCATCAGC |
| DDIT3 | GGAAACAGAGTGGTCATTCCC | CTGCTTGAGCCGTTCATTCTC |
| β-Actin | CACCATTGGCAATGAGCGGTTC | AGGTCTTTGCGGATGTCCACGT |

## CRISPR/Cas9

KO cell lines were generated by transfecting RPE1 cells with the plasmid pLentiCRISPRV2 (a gift from Feng Zhang, Addgene, #52961; RRID:Addgene_52961) containing the sgRNAs. Puromycin selection was started 24 hr after transfection and continued for 7 days after which the selection medium was replaced with normal culture medium, and the cells cultured for 2–3 days before diluting ~1 cell/100 μl in a 96-well plate. Wells with single clones were expanded and sgRNA target region sequenced. Western blotting was used to confirm the KO.

To generate DELE1 HA knock-in cells, a combination of three different plasmids were electroporated using the Neon electroporation system (Thermo Fisher) as follows: 10 μg eSpCas9(1.1)_No_FLAG_ATP1A1_G3_Dual_sgRNA (a gift from Yannick Doyon, Addgene, #86613; RRID:Addgene_86613) with the guide RNA specific for the target gene and ATP1A1, 10 μg pBM16A-backbone with a glycine and serine linker and 3×HA tag, flanked on either side by the homology arms mapping to around 250 bp upstream and downstream of the stop codon of the targeted gene and 10 μg pMK-ATP1A1_Q118RN129D plasmid as the selection marker. Electroporation was performed at a voltage of 1400 V with 1 pulse, 30 ms width on 5 million cells in resuspension buffer R for a 100 μL Neon tip. After electroporation, the cells were cultured for 3 days after which 100 μM ouabain (TargetMol) was added for 5 days. Single-cell clones were obtained by limiting dilution and clones analyzed by PCR and Sanger sequencing.

## ATF4 reporter assay

A 363 bp fragment of the ATF4 5' untranslated region (sequence TTTCTACTTTGCC…TTTGGGGG CTGA) was cloned before firefly luciferase driven by CMV promoter in pEZX-FR01 plasmid (Genecopoeia). For transfections, 50 ng ATF4 reporter+30 ng Flag-HRI or GFP-Flag+70 ng untagged PEBP1, GFP-Flag, or DELE1-ΔMTS overexpression plasmids were mixed with 0.15 μl Plus Reagent and 0.3 μl Lipofectamine LTX (Thermo Fisher) in OptiMEM medium and added to each well plated with 12,000 PEBP1 KO cells/well the day before. After 24 hr, dual luciferase assay was performed using Dual-Lumi reagents (RG088M, Beyotime, China). Luminescence values were normalized by dividing firefly luminescence values with Renilla luciferase signal and setting ATF4 reporter activity co-transfected with GFP as 1.

## NanoBiT protein interaction assay

Full-length ORFs of PEBP1 and EIF2S1 (eIF2α) were amplified from RPE1 cDNA, cloned into pcDNA3.1+vector and tagged C-terminally with SmBiT and LgBiT (*Dixon et al., 2016*), respectively. The LgBiT was synthesized at SynbioB (Tianjin, China). Point mutations were generated by standard site-directed mutagenesis and validated by sequencing. For transfection, 293T cells (180,000 cells/ml) were plated in white 96-well tissue culture plates (Biosharp, BS-MP-96W) in a total volume of 100 μl/well. The next day, 50 ng SmBiT and 50 ng of LgBiT expression constructs were mixed with 0.4 μl of 1 mg/ml polyethylenimine (PEI, Cat.No. 408727, Sigma-Aldrich) and added to each well. After 24 hr, cells were either left untreated or treated as indicated. Finally, 0.5 μl Nano-Glo Vivazine Live Cell substrate (Promega, N2581) diluted in a total volume of 10 μl DMEM was added to each well and luminescence measured with Spark Multimode Microplate Reader (Tecan).

For the real-time monitoring, Nano-Glo Vivazine Live Cell substrate was added in 1× final concentration in 90 μl complete medium supplemented with 25 mM HEPES, pH 7.3. Cells were incubated for 30–45 min at 37°C, before adding the drugs in 10 μl volume. Luminescence was monitored with 5 min interval at 37°C using SpectraMax iD5 (Molecular Devices). Data was normalized to the initial luminescence value after adding the drugs.

## Chemical compounds

The following small molecules were used. The concentrations used are indicated separately for each data figure.

| Compound | Supplier | Catalogue number | CAS number | Used for |
|---|---|---|---|---|
| Oligomycin A | SelleckChem | S1478 | 579-13-5 | Complex V inhibition |
| Bedaquiline | Aladdin | B413184 | 845533-86-0 | Complex V inhibition |
| Antimycin A | Abcam | AB141904 | 1397-94-0 | Complex III inhibition |
| Rotenone | Solarbio | R8130 | 83-79-4 | Complex I inhibition |
| 2-Deoxy-D-glucose | Macklin | D807272 | 154-17-6 | Glycolysis inhibition |
| CCCP | Solarbio | C6700 | 555-60-2 | ETC uncoupler |
| FCCP | TargetMol | T6834 | 370-86-5 | ETC uncoupler |
| Rosuvastatin | Solarbio | IR0150 | 147098-20-2 | Inhibitor of cholesterol synthesis |
| Mdivi-1 | Beyotime | SC8028 | 338967-87-6 | Inhibitor of mitochondrial division |
| PD 0332991 isethionate - Palbociclib | Sigma-Aldrich | PZ0199 | 827022-33-3 | Cdk4/6 inhibitor used as positive control for MS-CETSA |
| Tunicamycin | Cell Signaling Technology | 12819s | 11089-65-9 | Induces ER stress and eIF2a phosphorylation via PERK kinase |
| Puromycin | BBI-Life Sciences | A610593-0026 | 58-58-2 | Used for puromycin selection for stable cell lines |
| BAM15 | MedChemExpress | HY-110284 | 210302-17-3 | Electron transport chain uncoupler which does not depolarize plasma membrane |
| DFOM Deferoxamine | Sigma-Aldrich | D9533 | 138-14-7 | Chelates iron and induces ISR via HRI kinase |
| Poly(I:C) | MedChemExpress | HY-107202 | 24939-03-5 | Mimics viral infection and activates ISR via PKR kinase |
| Sodium (meta)arsenite | Sigma-Aldrich | S7400 | 7784-46-5 | Induces oxidative stress and induces ISR via HRI kinase |
| Trametinib | MedChemExpress | HY-10999 | 871700-17-3 | MEK inhibitor used to test if PEBP1-mediated effect on ISR is related to PEBP1 function as a Raf kinase inhibitor |
| Phorbol-12-myristate-13-acetate (PMA) | Rhawn | R038872 | 16561-29-8 | Induction of PEBP1 phosphorylation at Ser153 |

| Compound | Supplier | Catalogue number | CAS number | Used for |
|----------|----------|------------------|------------|----------|
| Sephin-1 | Sigma-Aldrich | SML1356 | 951441-04-6 | Inhibitor of GADD34 (PPP1R15A), the stress-inducible eIF2α phosphatase subunit |
| Raphin-1 | Aladdin | R287653 | 2022961-17-5 | Inhibitor of CREP (PPP1R15B), the constitutively expressed eIF2α phosphatase subunit |

## MS-CETSA/TPP

Three sub-confluent 10 cm dishes of 143B were treated with each of the compounds or DMSO for 30 min. All compounds were assayed in biological duplicates. Culture medium was removed and cells washed with ice-cold PBS. Each plate was lysed with 1 ml 1% NP40 (IGEPAL CA-630, Cat.No. I8896, Sigma-Aldrich) in PBS containing 1× proteinase inhibitor cocktail (Cat.No. P1008, Beyotime) and the respective drugs. Lysates were transferred to 1.5 ml tubes, centrifuged at 16 000 × $g$ for 10 min at 4°C and the lysates from the three plates were pooled. 800 µl of each lysate was snap-frozen in liquid nitrogen as total proteome to assess any protein level changes. The remaining lysate was divided into 130 µl aliquots in 12 thin-walled PCR tubes, heated in a gradient thermal cycler (Eppendorf Mastercycler X50s) for 7 min at temperatures ranging between 45–50°C (45.0, 45.8, 46.9, 48.0, 48.9, and 50.0°C) and 51–56°C (51.0, 51.8, 52.9, 54.0, 54.9, 56.0°C). Equal volumes from each temperature point were combined into the 45–50°C and 51–56°C pools and the precipitated proteins removed by centrifugation at 16,000 × $g$ for 10 min at 4°C. The soluble supernatants were frozen in liquid nitrogen and stored at –80°C until processed for mass spectrometry. Briefly, trypsin digested peptides were desalted and lyophilized, reconstituted in 0.5 M TEAB and labeled with TMTs. Pooled samples were run using Q Exactive Plus (Thermo Fisher). The MS/MS data were processed using MaxQuant (version 1.5.2.8). Tandem mass spectra were searched against human Uniprot database concatenated with reverse decoy peptides. The peptide search allowed up to four missing trypsin cleavages with mass tolerance for precursor ions 20 ppm in the first search and 5 ppm in the main search. Mass tolerance for fragment ions was 0.02 Da. FDR was adjusted to <1% and minimum score for modified peptides to 40. Sample preparation and proteomics was performed at PTM Biolabs (Hangzhou, China).

## In vitro protein thermal shift assay

100× SYPRO Orange Protein Gel Stain (Sigma-Aldrich, S5692) was mixed with 1 µM purified recombinant PEBP1 (Cat.No. 14582-HNAE, Sino Biological), and aliquoted it into each well of the 96-well plate in 11.5 µl total volume. 1 µl of oligomycin A were added to obtain the indicated final concentrations. Thermal melt curves were obtained using CFX96 Touch Real-Time PCR System (Bio-Rad) using the manufacturer's instructions (available here).

## RNAseq

RPE1 control KO and PEBP1 KO cells were seeded at the density of 0.6×10$^5$ cells/ml in a 10 cm dish. Cells were incubated at 37°C 5% CO$_2$ for 42 hr to reach ~90% confluence. Three dishes of control and PEBP1 KO cells were treated with 1 µM oligomycin A or equal volume of DMSO as control for 6 hr, washed twice with PBS and lysed in 1 ml TRIzol on ice. Lysates were frozen in liquid nitrogen and stored at –80°C. RNA library preparation and sequencing was performed at Novogene (Beijing). Briefly, sequencing libraries were generated from total RNA using NEBNext Ultra RNA Library Prep Kit for Illumina (NEB, Cat.No. E7530L) following the manufacturer's recommendations. After random fragmentation, 370–420 bp cDNA fragments were selected for paired end sequencing of 150 bp. After removal of poor-quality reads, the remaining reads were mapped against human transcriptome (Ensembl v109) using Salmon (version 1.3.0) to obtain transcript abundances. Gene level summaries were generated by *tximport*. Differential gene expression analysis for RPE1 cells (accession:

GSE247262) and bone marrow macrophage (accession: GSE264092) were performed using DESeq2 from variance stabilizing transformation (vst) normalized counts. Heatmap was drawn using *pheatmap* and *ggplot2* packages in R (R version 4.2.0). For functional profiling of RNAseq data, we analyzed the enrichment of differentially expressed genes (using p.adj<0.001 as the cut-off).

## Western blotting

143B or RPE1 cells (60,000 cells/ml) were plated in six-well plates (2 ml/well) a day before treatments as indicated in each figure legend. Cells were washed with PBS and lysed with 1% NP40-PBS for WB, containing protease and phosphatase inhibitors (Beyotime, Cat.No. P1051). Equal amount of total protein (typically 20 or 30 µg/sample) were separated either on 4–12% or 4–20% SurePAGE Bis-Tris gels (GenScript) and transferred on nitrocellulose membrane (Merck Millipore) using the eBlot L1 transfer system (GenScript). Membranes were blocked with QuickBlock Blocking Buffer (Beyotime, P0252), incubated with primary antibodies in QuickBlock Primary Antibody Dilution Buffer (Beyotime, P0256) or SignalUp Western Blot sensitizer (Beyotime, P0272), followed by HRP (CST, #7074) or AlexaFluor Plus 680/800 conjugated secondary antibodies (Invitrogen, A32735 and A32729). The signals were detected with Li-COR Odyssey imager and processed and using LI-COR ImageStudio software. Only blots probed with fluorescent secondary antibodies were used for quantitation.

For protein synthesis assay using HPG and click-chemistry, RPE1 cells (50,000 cells/ml) were plated in six-well plates (2 ml/well) and cultured for 48 hr. Cells were treated with oligomycin or tunicamycin for 4 hr, washed once with PBS, then incubated with 1 ml/well pre-warmed L-methionine-free DMEM (D0422, Sigma-Aldrich) supplemented with 200 µM L-cystine, 2 mM L-glutamine, and 10 mM HEPES for 30 min. Oligomycin or tunicamycin were maintained during this and the subsequent period. HPG (ST2057, Beyotime) was added to 50 µM final concentration and incubated for 2 hr, cells were washed twice with ice-cold PBS and lysed with 50 µl 1% NP40-PBS containing protease and phosphatase inhibitors. Equal amounts of total protein in 50 µl volume were mixed with 150 µl Click-reaction cocktail containing fluorescent Azide-647 (Beyotime, C0081) and incubated for 30 min at room temperature. SDS-PAGE sample buffer was added, and the samples heated and separated on a 4–20% SDS-PAGE gel and transferred on nitrocellulose membrane. Azide-647 was visualized with the 700 nm channel on the Li-COR Odyssey.

## Antibodies

The following antibodies were used:

| Antigen | Supplier | Cat.No. | Host species | Used dilution | Lot number | RRID |
|---|---|---|---|---|---|---|
| RKIP/PEBP1 | Abcam | ab76582 | Rabbit | 1:5000 | GR3257923-2 | AB_1267285 |
| PEBP1 Phospho-Ser153 | Diagbio | db13954 | Rabbit | 1:1000 | X0630P | AB_3371705 |
| Total OXPHOS Human WB Antibody Cocktail (to detect ATP5F1A) | Abcam | ab110411 | Mouse | 1:1000 | M5406 | AB_2756818 |
| P-eIF2a | Cell Signaling Technology | #3398 | Rabbit | 1:2000 | 8 | AB_2096481 |
| eIF2a | Cell Signaling Technology | #5324 | Rabbit | 1:2000 | 8 | AB_10692650 |
| ATF-4 | Cell Signaling Technology | #11815 | Rabbit | 1:1000 | 5 | AB_2616025 |
| b-Actin | GenScript | A00702 | Mouse | 1:3000 | 19G001892 | AB_914102 |
| HA | GenScript | A01244 | Mouse | 0.5 µg/ml | H2210004-A | AB_1289306 |
| CHOP | Proteintech | 15204-1-AP | Rabbit | 1:2000 | 00132291 | AB_2292610 |
| GADD34/PPP1R15A | Proteintech | 10449-1-AP | Rabbit | 1:2000 | 00102093 | AB_2168724 |

*Continued on next page*

*Continued*

| Antigen | Supplier | Cat.No. | Host species | Used dilution | Lot number | RRID |
|---|---|---|---|---|---|---|
| CREP/PPP1R15B | Proteintech | 14634-1-AP | Rabbit | 1:5000 | 00123911 | AB_2300036 |
| HRI | Proteintech | 20499-1-AP | Rabbit | 1:1000 | 00105271 | AB_10697665 |
| ERK1/2 | Proteintech | 11257-1-AP | Rabbit | 1:5000 | 00135728 | AB_2139822 |
| P-ERK1/2 | Proteintech | 28733-1-AP | Rabbit | 1:5000 | 00124649 | AB_2881202 |

## Quantitative RT-PCR

143B or RPE1 cells (60,000 cells/ml) were plated in six-well plates (2 ml/well) a day before treatment. Total RNA was extracted using FastPure cell/tissue total RNA isolation Kit (Vazyme). First-strand cDNA was synthesized using the HiScript Q select RT SuperMix for qPCR (Vazyme) and qPCR was performed with ChamQ Universal SYBR qPCR master Mix (Vazyme). Fold changes in expression were calculated using $2^{-\Delta\Delta Ct}$ method normalizing the Ct values to β-actin.

## Confocal microscopy

143B cells (40,000 cells/ml) were plated in 24-well plates a day before transfection. 25 nM siRNA was transfected using Lipofectamine 3000 (L3000-015, Invitrogen). After 48 hr, cells were reseeded into an eight-well chambered cover glass (Cellvis, Cat.No. C8-1.5H-N) (300 μl/well). Cells were incubated with 1 μM oligomycin A for 1 hr and stained with Hoechst 33342 (1:2000 dilution, Beyotime, C1022) and Mito-Tracker Red CMXRos (1:2000, Beyotime, C1049B) for 30 min. Images were captured with a spinning-disk confocal microscopy (Nikon) equipped with 60×/1.4 numerical aperture objective. Maximum intensity projection images were generated using Fiji (version 2.14.0/1.54f).

## Flow cytometry

143B cells were treated for 1 hr with 1 μM oligomycin A and 0.5 μM BAM15 as indicated and stained with tetramethylrhodamine ethyl ester (TMRE, 10 nM, Solarbio, T7910) for 30 min at 37°C. Cells were trypsinized, washed twice with HBSS, and analyzed with Cytek Aurora flow cytometer (Cytek Biosciences, Fremont, CA, USA). Data was processed with FlowJo software (BD Biosciences).

## Oxygen consumption rate

OCR and ECAR were measured using Seahorse XFe96 Analyzer (Agilent Technologies) following the manufacturer's instructions. RPE1 cells ($1\times10^4$) of the indicated genotypes were seeded in 96-well Seahorse XFe96 cell culture microplates (Agilent Technologies) for 24 hr in eight replicate wells. The culture medium was changed to Seahorse XF DMEM medium (Agilent Technologies, 103575-100) supplemented with 10 mM glucose, 1 mM pyruvate, and 2 mM L-glutamine 1 hr prior to the assay by washing once with this medium followed by addition of 180 μl medium per well. Mitochondrial stress test was performed following the standard protocol with final concentrations of 1.5 μM oligomycin A (Selleck, Cat.No. S1478), 1.5 μM FCCP (TargetMol, T6834), and 0.5 μM rotenone (Solarbio, R8130) plus antimycin A (Abcam, AB141904) (Rot/AA). To determine the cell numbers, 5 μg/ml Hoechst 33342 (Beyotime, C1022) was added into each well after the OCR measurement and fluorescence images of the wells were taken using High-Content Imaging System IXM-C (Molecular Devices). The calculated cell numbers were imported into the Seahorse Wave Controller software (version 2.6.3.8) for data normalization.

## AlphaFold-Multimer prediction

The potential complex between eIF2α and PEBP1 (UniProt IDs P05198 and P30086, respectively) was predicted from the amino acid sequences based on AlphaFold structure prediction model v2.3.2 implemented with ColabFold 1.5.3. at https://www.tamarind.bio/. The settings used were five models, MSA Mode: mmseqs2_uniref_env, 3 recycles without relaxations in an unpaired_paired mode without a template. The resulting 'top' model had pLDDT = 85.6, pTM = 0.501, and ipTM = 0.155.

## Statistical analysis

The proteome data was analyzed using AID, available from https://gygi.med.harvard.edu/software and https://github.com/alex-bio/TPP (*Panov, 2019*), and run locally using R environment. The protein abundances from the duplicate measurements were averaged for AID. This software ranks most likely shifted proteins by the logarithm of the Multivariate Normal p-value 'log_Multiv_Norm_pval', the descending 'pval_count' (i.e. in how many pools the protein shows p-value<0.05), and the decreasing magnitude of 'sum_of_signs' (i.e. thermal stabilization or destabilization). The AID method does not explicitly use log2 fold changes, but it does consider the relative abundance of proteins compared to all other proteins under different temperature fractions. Since the largest shift for each protein can be at any temperature, this combined pooling and multivariate approach indicates the proteins with most likely changes across the range of temperatures used.

For comparing responses in real-time monitoring of luminescence complementation assays, *trapz* function in *pracma* package (version 2.4.4) in R was used to calculate AUC. For statistical comparison between the AUCs in the control and drug-treated groups, two-sided t test was used. For comparisons with three or more groups, ANOVA followed by Tukey's post hoc test was used. All biological replicate samples were included in the analyses unless clear technical flaws were identified in the preparation of that sample. All experiments were repeated independently at least two to three times, except for proteomics and RNAseq.

## Acknowledgements

We thank Drs. Thomas Tan and David Tollervey for discussion, Chan Kuan Yoow for advice with knock-ins, Zhi Hong for manuscript comments, and the Core Facilities at the Zhejiang University-University of Edinburgh Institute for providing essential instrumentation. Materials generated in this study are available from the corresponding author upon reasonable request.This work was financially supported by National Science Foundation of China (grant number 31970706). IM and JC are supported by the dual award PhD degree program in Integrative Biomedical Sciences, while LC and YK are supported by the ZJU PhD degree program at Zhejiang University-University of Edinburgh Institute.

## Additional information

### Funding

| Funder | Grant reference number | Author |
|---|---|---|
| National Natural Science Foundation of China | 31970706 | Mikael Björklund |

The funders had no role in study design, data collection and interpretation, or the decision to submit the work for publication.

### Author contributions

Ling Cheng, Ian Meliala, Yidi Kong, Investigation; Jingyuan Chen, Formal analysis; Christopher G Proud, Conceptualization, Writing – review and editing; Mikael Björklund, Conceptualization, Formal analysis, Supervision, Funding acquisition, Visualization, Writing – original draft, Writing – review and editing

### Author ORCIDs

Ling Cheng https://orcid.org/0000-0003-0240-0385
Mikael Björklund https://orcid.org/0000-0002-2176-681X

Reviewer #1 (Public review): https://doi.org/10.7554/eLife.102852.2.sa1
Reviewer #2 (Public review): https://doi.org/10.7554/eLife.102852.2.sa2
Reviewer #3 (Public review): https://doi.org/10.7554/eLife.102852.2.sa3
Author response https://doi.org/10.7554/eLife.102852.2.sa4

# Additional files

## Supplementary files

MDAR checklist

Supplementary file 1. Proteomics data for the small molecules used for the MS-CETSA experiments.

## Data availability

RNAseq data has been deposited at the NCBI GEO repository (accession GSE247262). All other data can be found as supplementary material, including source data containing the original western blot images.

The following dataset was generated:

| Author(s) | Year | Dataset title | Dataset URL | Database and Identifier |
|---|---|---|---|---|
| Cheng L, Kong Y, Bjorklund M | 2024 | RNA-seq of control and PEBP1 knockout human RPE1 cells treated with or without oligomycin | https://www.ncbi.nlm.nih.gov/geo/query/acc.cgi?acc=GSE247262 | NCBI Gene Expression Omnibus, GSE247262 |

The following previously published datasets were used:

| Author(s) | Year | Dataset title | Dataset URL | Database and Identifier |
|---|---|---|---|---|
| Zhao F | 2024 | RKIP regulates bone homeostasis by mediating the differentiation fate of bone niche macrophages | https://www.ncbi.nlm.nih.gov/geo/query/acc.cgi?acc=GSE264092 | NCBI Gene Expression Omnibus, GSE264092 |
| Chen et al. | 2022 | MOSTA: Mouse Organogenesis Spatiotemporal Transcriptomic Atlas | https://db.cngb.org/stomics/mosta/spatial/ | Spatial Transcript Omics DataBase, stomics/mosta/spatial/ |
| Karlsson M, Zhang C, Méar L, Zhong W, Digre A, Katona B, Sjöstedt E, Butler L, Odeberg J, Dusart P, Edfors F, Oksvold P, von Feilitzen K, Zwahlen M, Arif M, Altay O, Li X, Ozcan M, Mardinoglu A, Fagerberg L, Mulder J, Luo Y, Ponten F, Uhlén M, Lindskog C | 2021 | A single-cell type transcriptomics map of human tissues | https://www.proteinatlas.org/ENSG00000089220-PEBP1/single+cell+type/liver | ProteinAtlas, ENSG00000089220-PEBP1/single+cell+type/liver |

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
