## [Editor Report · eLife Assessment]

In this article, Cheng et al present an **important** finding that advances the understanding of mitochondrial stress response(s). The authors employed mass spectrometry-based methods in conjunction with standard molecular and cellular biology techniques to provide **compelling** evidence that phosphatidylethanolamine-binding protein 1 (PEBP1) acts as a pivotal regulator of the mitochondrial component of integrated stress response. Notwithstanding that this discovery is likely to be of significant interest to researchers across a broad spectrum of disciplines ranging from cell biology to neuroscience, it was thought that further mechanistic dissection of the role of PEBP1 in modulating integrated stress response may further strengthen this study.

---

## [Referee Report · Reviewer #1 (Public review)]

Summary:

In this study, the authors use thermal proteome profiling to capture changes in protein stability following a brief (30 min) treatment of cells with various mitochondrial stressors. This approach identified PEBP1 as a potentiator of Integrated Stress Response (ISR) induction by various mitochondrial stressors, although the specific dynamics vary by stressor. PEBP1 deletion attenuates DELE1-HRI-mediated activation of the ISR, independent of its known role in the RAF/MEK/ERK pathway. These effects can be bypassed by HRI overexpression and do not affect DELE1 processing. Interestingly, in cells, PEBP1 physically interacts with eIF2alpha, but not its phosphorylated form (eIF2alpha-P), leading the authors to suggest that PEBP1 functions as a scaffold to promote eIF2alpha phosphorylation by HRI.

Strengths:

The authors present a clear and well-structured study, beginning with an original and unbiased approach that effectively addresses a novel question. The investigation of PEBP1 as a specific regulator of the DELE1-HRI signaling axis is particularly compelling, supported by extensive data from both genetic and pharmacological manipulations. Including careful titrations, time-course experiments, and orthogonal approaches strengthens the robustness of their findings and bolsters their central claims.

Moreover, the authors skillfully integrate publicly available datasets with their original experiments, reinforcing their conclusions' generality and broader relevance. This comprehensive combination of methodologies underscores the reliability and significance of the study's contributions to our understanding of stress signaling.

Weaknesses:

While the study presents exciting findings, there are a few areas that could benefit from further exploration. The HRI-DELE1 pathway was only recently discovered, leaving many unanswered questions. The observation that PEBP1 interacts with eIF2alpha, but not with its phosphorylated form, suggests a novel mechanism for regulating the Integrated Stress Response (ISR). However, as they note themselves, the authors do not delve into the biochemical or molecular mechanisms through which PEBP1 promotes HRI signaling. Given the availability of antibodies against phosphorylated HRI, it would have been interesting to explore whether PEBP1 influences HRI phosphorylation. Furthermore, since the authors already have recombinant PEBP1 protein (as shown in Figure 1D), additional in vitro experiments such as in vitro immunoprecipitation, FRET, or surface plasmon resonance (SPR) could have confirmed the interaction with eIF2alpha. Future studies might investigate whether PEBP1 directly interacts with HRI, stimulates its auto-phosphorylation or kinase activity, or serves as a template for oligomerization, potentially supported by structural characterization of the complex and mutational validation.

Another point of weakness is the unclear significance of the 1.5-2x enhanced interaction with eIF2alpha upon PEBP1 phosphorylation, as there is little evidence to show that this increase has any downstream effects. The ATF4-luciferase reporter experiments, comparing WT and S153D overexpression, may have reached saturation with WT, making it difficult to detect further stimulation by S153D. Additionally, expression levels for WT and mutant forms are not provided, making it challenging to interpret the results. It would also be interesting to explore whether combined mitochondrial stress and PMA treatment further enhance the ISR.

Lastly, while the authors claim that oligomycin does not significantly alter the melting temperature of recombinant PEBP1 in vitro, the data in Figure S1D suggest a small shift. Without variance measures across replicates or background subtraction, this claim is less convincing. The inclusion of statistical analyses would strengthen the interpretation of these results.

Impact on the field:

The study's relevance is underscored by the fact that overactive ISR is linked to a broad range of neurodegenerative diseases and cognitive disorders, a field actively being explored for therapeutic interventions, with several drugs currently in clinical trials. Similarly, mitochondrial dysfunction plays a well-established role in brain health and other diseases. Identifying new targets within these pathways, like PEBP1, could provide alternative therapeutic strategies for treating such conditions. Therefore, gaining a deeper understanding of the mechanisms through which PEBP1 influences ISR regulation is highly pertinent and could have far-reaching implications for the development of future therapies.

---

## [Referee Report · Reviewer #2 (Public review)]

Summary:

In this work, Cheng et al use the TPP/MS-CETSA strategy to discover new components for the mitochondria arm of the Integrated Stress Response. By using short exposures of several drugs that potentially induce mitochondrial stress, they find significant CETSA shifts for the scaffold protein PEBP1 both for antimycinA and oligomycin, making PEBP1 a candidate for mitochondrial-induced ISR signaling. After extensive follow-up work, they provide good support that PEBP1 is likely involved in ISR, and possibly act through an interaction with the key ISR effector node EIF2a.

Strengths:

The work adds an important understanding of ISR signaling where PEBP1 might also constitute a druggable node to attenuate cellular stress. Although CETSA has great potential for dissecting cellular pathways, there are few studies where this has been explored, particularly with such an extensive follow-up, also giving the work methodological implications. Together I therefore think this study could have a significant impact.

Weaknesses:

The TPP/MS-CETSA experiment is quite briefly described and might have a too relaxed cut-off. The assays confirming interactions between PEBP1 and EIF2a might not be fully conclusive.

---

## [Referee Report · Reviewer #3 (Public review)]

Summary:

In this paper, Chang and Meliala et al. demonstrate that PEBP1 is a modulator of the ISR, specifically through the induction of mitochondrial stress. The authors utilize thermal proteome profiling (TPP) by which they identify PEPB1 as a thermally stabilized protein upon oligomycin treatment, indicating its role in mitochondrial stress. Moreover, RNA-sequencing analysis indicated that PEBP1 may be specifically modulating the mitochondrial stress-induced ISR, as PEBP1 knock-out reduces phosphorylation of eIF2α. They also show that PEBP1 function is independent of ER stress specifically tunicamycin treatment and loss of PEBP1 does affect mitochondrial ISR but in an OMA1, DELE1 independent manner. Thus, the authors hypothesized that PEBP1 interacts directly with eIF2α, functioning as a scaffolding protein. However, direct co-immunoprecipitation failed to demonstrate PEBP1 and eIF2α potential interaction. The authors then used a NanoBiT luminescence complementation assay to show the PEBP1-eIF2a interaction and its disruption by S51 phosphorylation.

Strengths:

Taken together, this work is novel, and the data presented suggests PEBP1 has a role as a modulator of the mitochondrial ISR, enhancing the signal to elicit the necessary response.

Weaknesses:

The one major issue of this work is the lack of a mechanism showing precisely how PEBP1 amplifies the mitochondrial integrated stress response. The work, as it is described, presents data suggesting PEBP1's role in the ISR but fails to present a more conclusive mechanism.

---

## [Author Response]

We thank all the reviewers for their insightful comments on this work.

**Response to Reviewer #1:**

We greatly appreciate your comments on the general reliability and significance of our work. We fully agree that it would have been ideal to have additional evidence related to the role of PEBP1 in HRI activation. Unfortunately, we have not been able to find phospho-HRI antibodies that work reliably. The literature seems to agree with this as a band shift using total-HRI antibodies is usually used to study HRI activation. However, with the cell lines showing the most robust effect with PEBP1 knockout or knockdown, we are yet to convince ourselves with the band shifts we see. This could be addressed by optimizing phos-tag gels although these gels can be a bit tricky with complex samples such as cell lysates which contain many phosphoproteins.

To address the interaction between PEBP1 and eIF2alpha more rigorously we were inspired by the insights you and reviewer #2 provided. While we are unable to do further experiments, we now think it would indeed be possible to do this with either using the purified proteins and/or CETSA WB. These experiments could also provide further evidence for the role of PEBP1 phosphorylation. Although phosphorylation of PEBP1 at S153 has been implicated as being important for other functions of PEBP1, we are not sure about its role here. It may indeed have little relevance for ISR signalling.

For the in vitro thermal shift assay, we have performed two independent experiments. While it appears that there is a slight destabilization of PEBP1 by oligomycin, the ultimate conclusion of this experiment remains incomplete as there could be alternative explanations despite the apparent simplicity of the assay due the fluorescence background by oligomycin only. We now provide a lysate based CETSA analysis which does not display the same PEBP1 stabilization as the intact cell experiment. As for the signal saturation in ATF4-luciferase reporter assay, this is a valid point.

**Response to Reviewer #2:**

We strongly agree that CETSA has a lot of potential to inform us about cellular state changes and this was indeed the starting point for this project. We apologize for being (too) brief with the explanations of the TPP/MS-CETSA approach and we have now added a bit more detail. With regard to the cut-offs used for the mass spectrometry analysis, you are absolutely right that we did not establish a stringent cut-off that would show the specificity of each drug treatment. Our take on the data was that using the p values (and ignoring the fold-changes) of individual protein changes as in Fig 1D, we can see that mitochondrial perturbations display a coordinated response. We now realize that the downside of this representation is that it obscures the largest and specific drug effects. As mentioned in the response to Reviewer #1, we now also think that it would be possible to obtain more evidence for the potential interaction between PEBP1 and eIF2alpha using CETSA-based assays.

**Response to Reviewer #3:**

Thank you for your assessment, we agree that this manuscript would have been made much stronger by having clearer mechanistic insights. As mentioned in the responses to other reviewers above, we aim to address this limitation in part by looking at the putative interaction between PEBP1 and eIF2alpha with orthogonal approaches. However, we do realize that analysis of protein-protein interactions can be notoriously challenging due to false negative and false positive findings. As with any scientific endeavor, we will keep in mind alternative explanations to the observations, which could eventually provide that cohesive model explaining how precisely PEBP1, directly or indirectly, influences ISR signalling.

**Recommendations for the authors:**

**Reviewer #1 (Recommendations for the authors):**
The data overall are very solid, and I would only recommend the following minor changes:(1) Line 187 and line 268: there is perhaps a trend towards slightly increased ATF4-luc reporter with PEBP1-S153D, but it is not statistically significant, so I would tone down the wording here.

We now modified this part to "This data is consistent with the modest increase…" .

(2) The recently discovered SIFI complex (Haakonsen 2024, https://doi.org/10.1038/s41586023-06985-7) regulates both HRI and DELE1 through bifunctional localization/degron motifs. It seems like PEBP1 also contains such a motif, which suggests a potential mechanism for enrichment near mitochondria, perhaps even in response to stress. Maybe the authors could further speculate on this in the discussion.

While working on the manuscript, we considered the possibility that PEBP1 function could be related to SIFI complex and concluded that here is a critical difference: while SIFI specifically acts to turn off stress response signalling, loss of PEBP1 prevents eIF2alpha phosphorylation. We did not however consider that PEBP1 could have a localization/degron motif. Motif analysis by deepmito (busca.biocomp.unibo.it) and similar tools did not identify any conventional mitochondrial targeting signal although we acknowledge that PEBP1 has a terminal alpha-helix which was identified for SIFI complex recognition. We are not sure why you think PEBP1 contains such a motif and therefore are hesitant to speculate on this further in the manuscript.

(3) Line 358: references 50 and 45 are identical.

Thank you for spotting this. Corrected now.

(4) Figure S1D: it looks like Oligomycin has a significant background fluorescence, which makes interpretation of these graphs difficult - do you have measurements of the compound alone that can be used to subtract this background from the data? Based on the Tm I would say it does stabilize recombinant PEBP1, and there is no quantification of the variance across the 3 replicates to say there is no difference.

You are right, this assay is problematic due to the background fluorescence. The measurements with oligomycin only and subtracting this background results in slightly negative values and nonsensical thermal shift curves. We now additionally show quantification from two different experiments (unfortunately we ran out of reagents for further experiments), and this quantification shows that if anything, oligomycin causes mild destabilization of recombinant PEBP1. We also used lysate CETSA assay which does not show thermal stabilization of PEBP1 by oligomycin, ruling out a direct effect. We attempted to use ferrostatin1 as a positive control as it may bind PEBP1-ALOX protein complex, and it appeared to show marginal stabilization of PEBP1.

**Reviewer #2 (Recommendations for the authors):**
I have a few comments for the authors to address:(1) The MS-CETSA experiment is quite briefly described and this could be expanded somewhat. Not clear if multiple biological replicates are used. Is there any cutoff in data analysis based on fold change size (which correlated to the significance of cellular effects), etc? As expected from only one early timepoint (see eg PMID: 38328090), there appear to be a limited number of significant shifts over the background (as judged from Figure S1A). In the Excel result file, however (if I read it right) there are large numbers of proteins that are assigned as stabilized or destabilized. This might be to mark the direction of potential shifts, but considering that most of these are likely not hits, this labeling could give a false impression. Could be good to revisit this and have a column for what could be considered significant hits, where a fold change cutoff could help in selecting the most biologically relevant hits. This would allow Figure 1D to be made crisper when it likely dramatically overestimates the overlap between significant CETSA shifts for these drugs.

Fair point, while we focused more on PEBP1, it is important to have sufficient description of the methods. We used duplicate samples for the MS, which is probably the most important point which was absent from the original submission as is now added to the methods. We also added slightly more description on the data analysis. While the AID method does not explicitly use log2 fold changes, it does consider the relative abundance of proteins under different temperature fractions. Since the Tm (melting temperature) for each protein can be at any temperature, we felt that if would be complicated to compare fractions where the protein stability is changed the most and even more so if we consider both significance and log2FC. Therefore, we used this multivariate approach which indicates the proteins with most likely changes across the range of temperatures. To acknowledge that most of the statistically significant changes are not the much over the background as you correctly pointed out, we now add to the main text that “However, most of these changes are relatively small. To focus our analysis on the most significant and biologically relevant changes…” We also agree that it may be confusing that the AID output reports de/stabilization direction for all proteins. In general, we are not big fans of cutoffs as these are always arbitrary, but with multivariate p value of 0.1 it becomes clear that there are only a relatively small number of hits with larger changes. We have now added to the guide in the data sheet that "Primarily, use the adjusted p value of the log10 Multivariate normal pvalue for selecting the overall statistically significant hits (p<0.05 equals -1.30 or smaller; p<0.01 equals -2 or smaller)". We have also added to the guide part of the table that “Note that this prediction does not consider whether the change is significant or not, it only shows the direction of change”

(2) On page 4 the authors state "We reasoned that thermal stability of proteins might be particularly interesting in the context of mitochondrial metabolism as temperature-sensitive fluorescent probes suggest that mitochondrial temperature in metabolically active cells is close to 50{degree sign}C". I don't see the relevance of this statement as an argument for using TPP/CETSA. When this is also not further addressed in the work, it could be deleted.

Deleted. We agree, while this is an interesting point, it is not that relevant in this paper.

(3) To exclude direct drug binding to PEBP1, a thermofluor experiment is performed (Fig S1D). However, the experiment gives a high background at the lower temperatures and it could be argued that this is due to the flouroprobe binding to a hydrophobic pocket of the protein, and that oligomycin at higher concentrations competes with this binding, attenuating fluorescence. These are complex experiments and there could be other explanations, but the authors should address this. An alternative means to provide support for non-binding would be a lysate CETSA experiment, with very short (1-3 minutes) drug exposure before heating. This would typically give a shift when the protein is indicated to be CETSA responsive as in this case.

Agree. However, we don't have good means to perform the thermofluor experiments to rule out alternative explanations. What we can say is (as discussed above for reviewer #1, point 4) that quantification from two different experiments shows that oligomycin is does not thermally stabilizing recombinant PEBP1. To complement this conclusion, we used lysate CETSA assay which does not show thermal stabilization of PEBP1 by oligomycin. In this assay we attempted to use ferrostatin1 as a positive control as it may bind PEBP1-ALOX protein complex, and it appeared to show marginal stabilization of PEBP1. But since we lack a robust positive control for these assays, some doubt will inevitably remain.

(4) The authors appear to have missed that there is already a MS-CETSA study in the literature on oligomycin, from Sun et al (PMID: 30925293). Although this data is from a different cell line and at a slightly longer drug treatment and is primarily used to access intracellular effects of decreased ATP levels induced by oligomycin, the authors should refer to this data and maybe address similarities if any.

Apologies for the oversight, the oligomycin data from this paper eluded us at it was mainly presented in the supplementary data. We compared the two datasets and find found some overlap despite the differences in the experimental details. Both datasets share translational components (e.g. EIF6 and ribosomal proteins), but most notably our other top hit BANF1 which we mentioned in the main text was also identified by Sun et al. We have updated the manuscript text as "Other proteins affected by oligomycin included BANF1, which binds DNA in an ATP dependent manner [16], and has also identified as an oligomycin stabilized protein in a previous MS-CETA experiment [23]", citing the Sun et al paper.

(5) The confirmation of protein-protein interaction is notoriously prone to false positives. The authors need to use overexpression and a sensitive reporter to get positive data but collect additional data using mutants which provide further support. Typically, this would be enough to confirm an interaction in the literature, although some doubt easily lingers. When the authors already have a stringent in-cell interaction assay for PEBP1 in the CETSA thermal shift, it would be very elegant to also apply the CETSA WB assay to the overexpressed constructs and demonstrate differences in the response of oligomycin, including the mutants. I am not sure this is feasible but it should be straightforward to test.

This is a very good suggestion. Unfortunately, due to the time constraints of the graduate students (who must write up their thesis very soon), we are not able to perform and repeat such experiments to the level of confidence that we would like.

(6) At places the story could be hard to follow, partly due to the frequent introduction of new compounds, with not always well-stated rationale. It could be useful to have a table also in the main manuscript with all the compounds used, with the rationale for their use stated. Although some of the cellular pathways addressed are shown in miniatures in figures, it could be useful to have an introduction figure for the known ISR pathways, at least in the supplement. There are also a number of typos to correct.

We agree that there are many compounds used. We have attempted to clarify their use by adding this information into the table of used compounds in the methods and adding an overall schematic to Fig S1G and a note on line 132 "(see Figure 1-figure supplement 1G for summary of drugs used to target PEBP1 and ISR in this manuscript). We have also attempted to remove typos as far as possible.

(7) EIF2a phosphorylation in S1E does not appear to be more significant for Sodium Arsenite argued to be a positive control, than CCCP, which is argued to be negative. Maybe enough with one positive control in this figure?

This experiment was used as a justification for our 30 min time point for the proteomics. By showing the 30 min and 4 h time points as Fig 1G and Figure 1-figure supplement 1F, our point was to demonstrate that the kinetics of phosphorylation and dephosphorylation are relevant. As you correctly pointed out, the stress response induced by sodium arsenite, but also tunicamycin is already attenuated at the 4h time point. We prefer to keep all samples to facilitate comparisons.

(8) Page 7 reference to Figure S2H, which doesn't exist. Should be S3H.

Apologies for the mistake, now corrected to Figure 2-figure supplement 1B.

(9) Finally, although the TPP labeling of the method is used widely in the literature this is CETSA with MS detection and MS-CETSA is a better term. This is about thermal shifts of individual proteins which is a very well-established biophysical concept. In contrast, the term Thermal Proteome Profiling does not relate to any biophysical concept, or real cell biology concept, as far as I can see, and is a partly misguided term.

We changed the term TPP into MS-CETSA, but also include the term TPP in the introduction to facilitate finding this paper by people using the TPP term.

**Reviewer #3 (Recommendations for the authors):**
Major Issues(1) The one major issue of this work is the lack of a mechanism showing precisely how PEBP1 amplifies the mitochondrial integrated stress response. The work, as it is described, presents data suggesting PEBP1's role in the ISR but fails to present a more conclusive mechanism. The idea of mitochondrial stress causing PEBP1 to bind to eIF2a, amplifying ISR is somewhat vague. Thus, the lack of a more defined model considerably weakens the argument, as the data is largely corollary, showing KO and modulation of PEBP1 definitely has a unique effect on the ISR, however, it is not conclusive proof of what the authors claim. While KO of PEBP1 diminishes the phosphorylation of eIF2a, taken together with the binding to eIF2a, different pathways could be simultaneously activated, and it seems premature to surmise that PEBP1 is specific to mitochondrial stress. Could PEBP1 be reacting to decreased ATP? Release of a protein from the mitochondria in response to stress? Is PEBP1's primary role as a modulator of the ISR, or does it have a role in non-stress-related translation? A cohesive model would tie together these separate indirect findings and constitute a considerable discovery for the ISR field, and the mitochondrial stress field.

Thank you for your assessment, we agree that this manuscript would have been much stronger by having clearer mechanistic insights. As with any scientific endeavor, we will keep in mind alternative explanations to the observations, which could eventually provide that cohesive model explaining how precisely PEBP1, directly or indirectly, influences ISR signalling.

(2) The data relies on the initial identification of PEBP1 thermal stabilization concomitant with mitochondrial ISR induction post-treatment of several small molecules. However, the experiment was performed using a single timepoint of 30 minutes. There was no specific rationale for the choice of this time point for the thermal proteome profiling.

The reasoning for this was explicitly stated: "We reasoned that treating intact cells with the drugs for only 30 min would allow us to observe rapid and direct effects related to metabolic flux and/or signaling related to mitochondrial dysfunction in the absence of major changes in protein expression levels.”

Minor Issues(1) In Lines 163-166 the authors state "The cells from Pebp1 KO animals displayed reduced expression of common ISR genes (Figure 2F), despite upregulation of unfolded protein response genes Ern1 (Ire1α) and Atf6 genes. This gene expression data therefore suggests that Pebp1 knockout in vivo suppresses induction of the ISR". This statement should be reassessed. While an arm of the UPR does stimulate ISR, this arm is controlled by PERK, and canonically IRE1 and ATF6 do not typically activate the ISR, thus their upregulation is likely unrelated to ISR activation and does not contribute the evidence necessary for this statement.

Apologies for the confusion, we aimed to highlight that as there is an increase in the two UPR arms, it is more likely that ISR instead of UPR is reduced. We have now changed the statement to the following:

"The cells from Pebp1 PEBP1 KO animals displayed reduced expression of common ISR genes (Figure 2F), while there was mild upregulation of the unfolded protein response genes Ern1 (Ire1α) and Atf6 genes. This gene expression data therefore suggests that the reduced expression of common ISR genes is less likely to be mediated by changes in PERK, the third UPR arm, and more likely due to suppression of ISR by Pebp1 knockout in vivo."

(2) In Lines 169 and 170 the authors state "Western blotting indicated reduced phosphorylation of eIF2α in RPE1 cells lacking PEBP1, suggesting that PEBP1 is involved in regulating ISR signaling between mitochondria and eIF2α". This conclusion is not supported by evidence. A number of pathways could be activated in these knockout cells, and simply observing an increase in p-eIF2α after knocking out PEBP1 does not constitute an interaction, as correlation doesn't mean causation. This KO could indirectly affect the ISR, with PEBP1 having no role in the ISR. While taken together there is enough circumstantial evidence in the manuscript to suggest a role for PEBP1 in the ISR, statements such as these have to be revised so as not to overreach the conclusions that can be achieved from the data, especially with no discernible mechanism.

We have now revised this statement by removing the conclusion and stating only the observation: "Western blotting indicated reduced phosphorylation of eIF2α in RPE1 cells lacking PEBP1 (Fig. 3A)."